# Image processing tools for petabyte-scale light sheet microscopy data

Xiongtao Ruan [1] ✉, Matthew Mueller [1,2], Gaoxiang Liu[1], Frederik Görlitz[1,8], Tian-Ming Fu [3,9], Daniel E. Milkie [3], Joshua L. Lillvis[3], Alexander Kuhn[4], Johnny Gan Chong [1], Jason Li Hong [1], Chu Yi Aaron Herr[1], Wilmene Hercule[1], Marc Nienhaus[4], Alison N. Killilea [1], Eric Betzig [1,2,3,5,10] ✉ & Srigokul Upadhyayula [1,6,7,10] ✉

Light sheet microscopy is a powerful technique for high-speed three-dimensional imaging of subcellular dynamics and large biological specimens. However, it often generates datasets ranging from hundreds of gigabytes to petabytes in size for a single experiment. Conventional computational tools process such images far slower than the time to acquire them and often fail outright due to memory limitations. To address these challenges, we present PetaKit5D, a scalable software solution for efficient petabyte-scale light sheet image processing. This software incorporates a suite of commonly used processing tools that are optimized for memory and performance. Notable advancements include rapid image readers and writers, fast and memory-efficient geometric transformations, high-performance Richardson–Lucy deconvolution and scalable Zarr-based stitching. These features outperform state-of-the-art methods by over one order of magnitude, enabling the processing of petabyte-scale image data at the full teravoxel rates of modern imaging cameras. The software opens new avenues for biological discoveries through large-scale imaging experiments.

Light sheet microscopy enables fast three-dimensional (3D) imaging of cells, tissues and organs[1]. Within this realm, variants like multi-view selective plane illumination microscopy[2–4], lattice light sheet microscopy (LLSM)[5], axially swept light sheet microscopy[6,7] and single objective light sheet microscopy[8–10], offer higher resolution and fast imaging speed. Combined with expansion microscopy[11], these techniques have been used to image millimeter-scale or larger cleared and expanded specimens[12,13], while achieving nanoscale resolution. In such cases, the data produced from a single experiment can explode to the petabyte range. These high data generation rates introduce substantial challenges for data storage and processing that complicate visualization, assessment and analysis. First, even individual volumes from a four-dimensional (4D) time series can be so large as to render their preprocessing unwieldy or impossible for conventional processing codes. Second, acquisition in a non-Cartesian coordinate space adds substantial computational overhead. Third, light sheet data are often acquired at multi-terabyte-per-hour rates, which are too fast for conventional tools to process in real time. This impedes the rapid feedback needed to adjust imaging conditions or locations on the fly or to extract biological insights from the resulting datasets in a timely manner.

Numerous computational tools encompassing various functionalities have been developed to facilitate light sheet image preprocessing,

[1]Department of Molecular and Cell Biology, University of California, Berkeley, Berkeley, CA, US. [2]Howard Hughes Medical Institute, Berkeley, CA, US. [3]Janelia Research Campus, Howard Hughes Medical Institute, Ashburn, VA, US. [4]NVIDIA, Berlin, Germany. [5]Department of Physics, Helen Wills Neuroscience Institute, University of California, Berkeley, Berkeley, CA, US. [6]Molecular Biophysics and Integrated Bioimaging Division, Lawrence Berkeley National Laboratory, Berkeley, CA, US. [7]Chan Zuckerberg Biohub, San Francisco, CA, US. [8]Present address: Department of Microsystems Engineering, University of Freiburg, Freiburg, Germany. [9]Present address: Department of Electrical and Computer Engineering, Princeton University, Princeton, NJ, US. [10]These authors contributed equally: Eric Betzig, Srigokul Upadhyayula. ✉e-mail: xruan@berkeley.edu; betzige@janelia.hhmi.org; sup@berkeley.edu

including deskew and rotation[14,15], deconvolution[16,17], stitching[18,19] and visualization[20,21]. While these tools have shown to be valuable for light sheet images on the gigabyte scale, their utility wanes for data sizes surpassing the terabyte threshold, due to a lack of scalability and efficiency required to process images in real time. Furthermore, many of these tools are standalone applications, providing only partial processing steps in a specialized context and varying input formats and requirements. This situation often requires extensive manual effort to integrate them into multi-step workflows, limiting their utility, especially for large-scale data.

To address these challenges, particularly for long-term imaging of subcellular dynamics or vast multicellular image volumes, we developed PetaKit5D, a software solution designed to enable real-time processing of petabyte-scale imaging data. The software contains commonly used preprocessing and post-processing tools that are optimized for memory and performance, including deskew, rotation, deconvolution and stitching, all integrated into a high-performance computing framework capable of executing user-defined functions in a scalable and distributed manner.

To further increase throughput, we developed new algorithms for image input/output using the Zarr data format for image storage[22] and processing in conjunction with custom parallelized image readers and writers. These capabilities have been optimized for partitioned parallel processing of petabyte-scale datasets. The software incorporates an online mode during image acquisition to automatically process data and provide near-instantaneous feedback that is critical during long-term time series or high-throughput large sample imaging.

We developed PetaKit5D in MATLAB and offer Python wrappers for the deployed version. To ensure accessibility for users with little or no programming experience, the software includes a user-friendly graphical user interface (GUI).

## Results

### Overall design: distributed image processing framework

High frame rate modern cameras enable light sheet microscopes to capture images at nearly four terabytes per hour per camera. This presents formidable challenges for sustained image acquisition, real-time (de) compression, storage and processing, especially when using a single conventional workstation. In response, we developed a distributed computing architecture comprising a cluster of computing nodes and networked data storage servers that enables uninterrupted streaming and real-time processing of vast quantities of data continuously acquired over extended periods. Our standard workflow is illustrated in Fig. 1a.

Our approach uses a generic distributed computing framework in MATLAB to parallelize user-defined functions (Fig. 1b). The complete dataset or set of tasks is divided into distinct, self-contained subtasks, each appropriately sized for processing by an individual worker unit with one or more central processing unit (CPU) cores or GPUs (Fig. 1c,d). A conductor job orchestrates all operations, distributes tasks across the computing cluster and monitors their progress to completion (Fig. 1b). Failed jobs are automatically resubmitted with additional resources. Our MATLAB-based framework offers greater flexibility for various task types, enhanced robustness against failures and seamless integration across multiple processing steps, compared with Spark[23] and Dask[24]. We use it to manage all processing methods in PetaKit5D (Fig. 1e).

### Fast image readers and writers

Efficient image reading and writing are essential for real-time image processing. The widely used Tiff format stores two-dimensional (2D) and 3D microscopy data, offers the ability to compress (such as Lempel–Ziv–Welch (LZW) compression), and can include specialized metadata (such as Open Microscopy Environment TIFF[25]). Unfortunately, conventional image readers and writers for the Tiff format

are not designed for large-scale compressed data, being restricted to single-threaded operations (for example, libtiff). For instance, an 86-GiB 16-bit Tiff file (512 × 1,800 × 50,000) with libtiff (LZW compression) takes approximately 8.5 and 16 min to read and write, respectively (Extended Data Fig. 1a,b). These speeds pose a considerable bottleneck for efficient image processing, especially when the entire image needs to be loaded for the processing. Memory mapping is an alternative technique to process large images (for example, tifffile[26] in Python), but it is mostly limited to working with uncompressed data in their native byte order. This limitation can drastically increase storage requirements for large datasets, and the processing may still be constrained by slow read and write speeds.

To rectify this, we developed an optimized Tiff reader and writer in C++. This implementation leverages the OpenMP framework[27] to facilitate concurrent multi-threaded reading and writing (where only the compression process is parallelized in writing). Our Cpp-Tiff reader and writer are over 22 times and 7 times faster than conventional ones, respectively, for compressed data (Fig. 2a,b and Extended Data Fig. 1a,b for a 24-core node). Moreover, they substantially outperform the fast Python reader and writer library for Tiff files (tifffile[26] in Python; Fig. 2a,b and Extended Data Fig. 1a,b). Their speeds also increase linearly as more CPU cores are devoted to read/write operations (Extended Data Fig. 1e,f).

Although the Tiff format is commonly used for raw microscopy images, it is not the most efficient for parallel reading and writing, especially for very large image datasets. One major limitation is its single-container structure for file writes, which restricts it to single-threaded operations. To overcome this, we instead chose Zarr[22], a next-generation file format optimized for multi-dimensional data. Zarr efficiently stores data in nonoverlapping chunks of uniform size (border regions may be padded to match the full chunk size) and saves them as individual files. The format is similar to N5 (ref. 28), OME-Zarr[29] and TensorStore[30].

Zarr allows individual jobs to access only the specific region of interest at a given time. Distinct regions can be saved to disk independently and in parallel. Using optimized C/C++ code that leverages OpenMP, our Zarr reader/writer is 10–23 times faster for reading and 5–8 times faster for writing (Fig. 2c,d and Extended Data Fig. 1c,d) than the conventional implementation (using MATLAB's 'blockedImage' function to interface with the Python version of Zarr). Their performances also scale as more CPU cores are devoted to read/write operations (Extended Data Fig. 1g,h). Our implementation is also 5–10 and 5–8 times faster for read/write compared with the native Python implementation of Zarr (Fig. 2c,d and Extended Data Fig. 1c,d). Moreover, compared to TensorStore, Cpp-Zarr is 2.2 times and 1.5 times faster for reading and writing, respectively, for their preferred data orders (row-major in TensorStore and column-major in Cpp-Zarr; Supplementary Table 1). We opted to use the zstd compression algorithm[31] at compression level 1 to achieve better compression ratios at comparable read/write times to the lz4 algorithm[32] at level 5 (default in native and OME-Zarr; Extended Data Fig. 1i–k). We also created a Parallel Fiji Visualizer plugin that quickly opens compressed Tiff and Zarr files using our fast readers, enabling efficient data visualization and inspection in Fiji[33] (Supplementary Note 3).

### Fast combined deskew and rotation

In many light sheet microscopes, the excitation and detection objectives are oriented at an angle with respect to the substrate holding the specimen. It is convenient in such cases to image the specimen by sweeping it in the plane of this substrate, but the resulting raw image stack is then sheared and rotated with respect to the conventional specimen Cartesian coordinates (Fig. 3a). Traditionally, the data are transformed back to these coordinates by deskewing and rotating in two sequential steps (Fig. 3b). However, zero padding during deskew drastically increases data size, slowing computation and

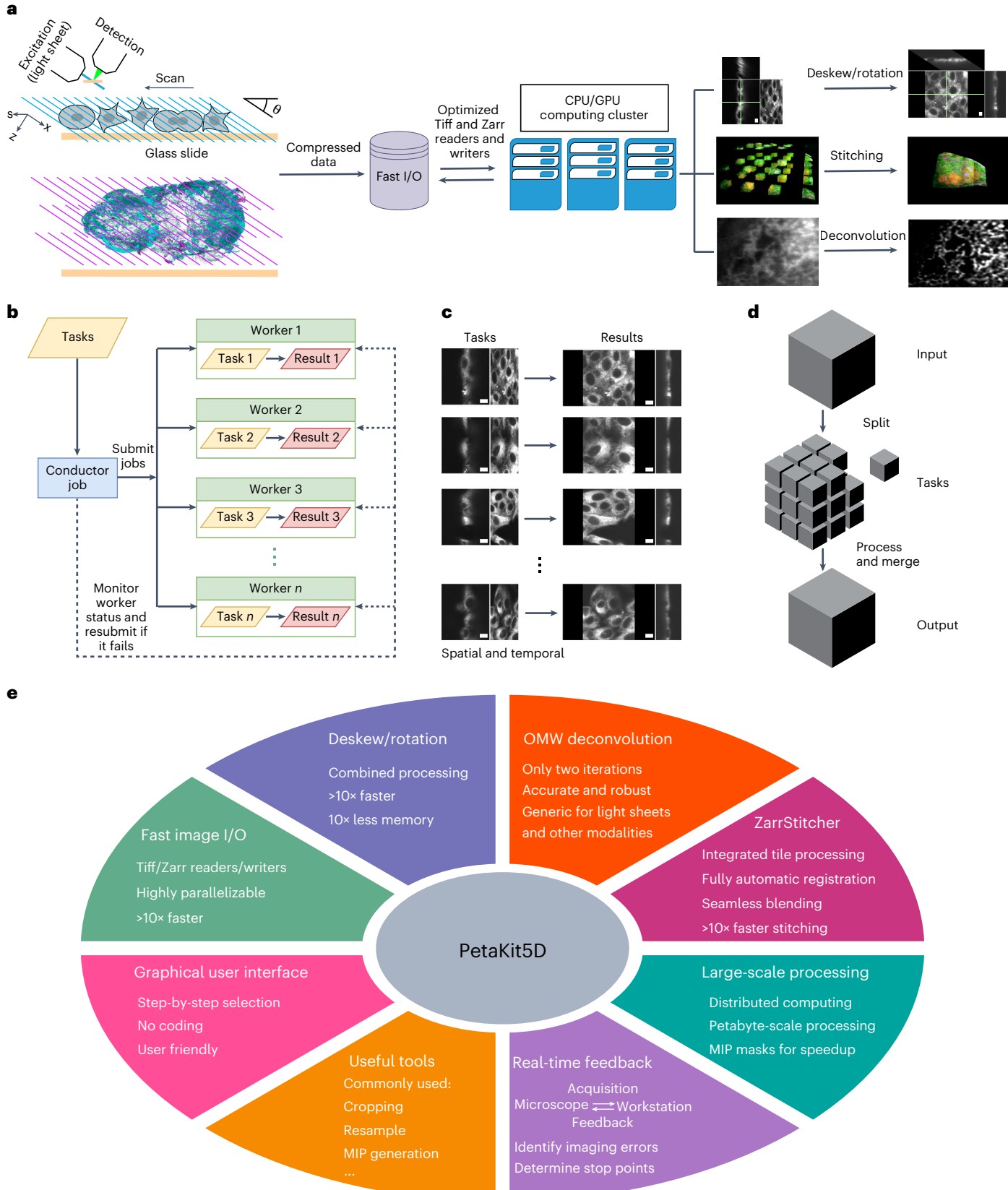

**Fig. 1 | Overall design of the image processing framework. a**, Image acquisition and processing workflows. **b**, Illustration of the generic distributed computing framework. **c**, Illustration of distributed processing of many independent files across multiple workers. **d**, Illustration of distributed processing of the split-process-merge mechanism for the distributed processing of a large image file. **e**, Overall functionalities and features in PetaKit5D.

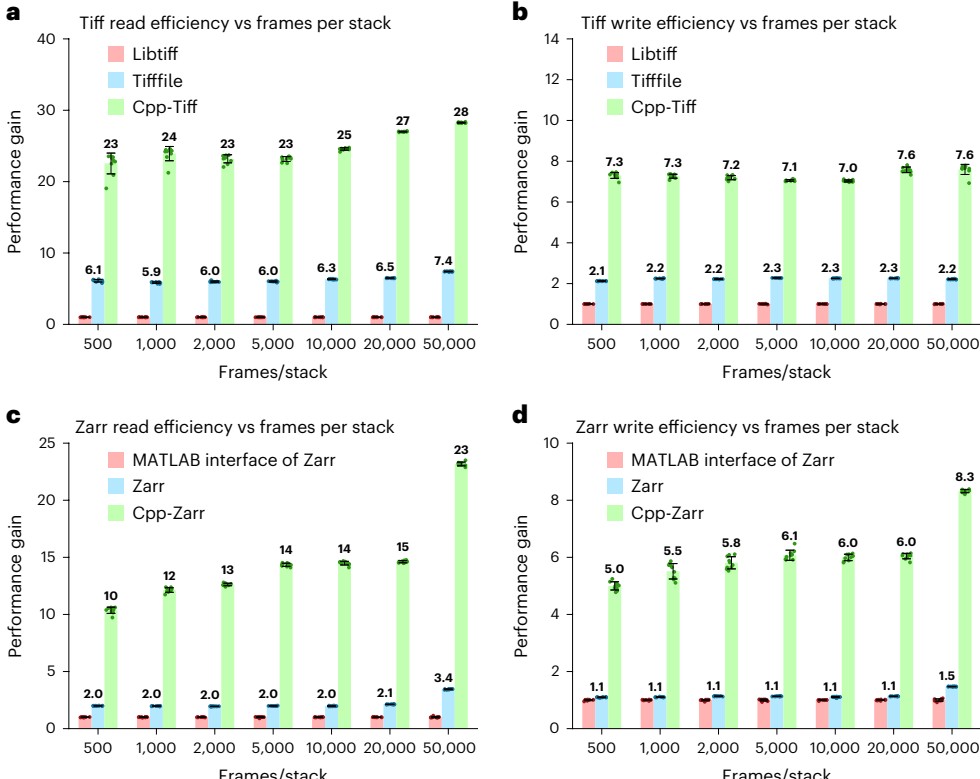

**Fig. 2 | Performance improvement factors of our C++ Tiff and Zarr readers and writers. a**, Performance gains of our Cpp-Tiff reader versus the conventional Tiff reader in MATLAB and tifffile reader in Python versus the number of frames in 3D stacks. **b**, Performance gains of Cpp-Tiff writer versus the conventional Tiff writer in MATLAB and the tifffile writer in Python versus the number of frames in 3D stacks. **c**, Performance gains of our Cpp-Zarr reader versus the conventional Zarr reader (MATLAB interface of Zarr) and native Zarr in Python versus the number of frames in 3D stacks. **d**, Performance gains of Cpp-Zarr writers versus the conventional Zarr writer (MATLAB interface of Zarr) and native Zarr in Python versus the number of frames in 3D stacks. The images have a uint16 frame size of 512 × 1,800 (*xy*) in all cases. The benchmarks were run independently ten times on a 24-core CPU computing node (dual Intel Xeon Gold 6146 CPUs). Data are shown as the mean ± s.d.

risking out-of-memory faults, particularly for large images with many frames (Fig. 3c).

To address this issue, we combined deskew and rotation into a single step, which is possible given that both operations are rigid geometric transformations. While prior studies have explored combining processing techniques using vertical interpolation in the deskew/rotated space and customized transformations[9,10,34–36], they are limited either in speed or the amount of data they can handle. Unlike previous approaches, our method first interpolates the data in the raw skewed space (depending on the scan step size), followed by standard affine transformation. When the ratio of the scan step size in the *xy* plane to the *xy* voxel size (defined as 'skew factor') is smaller than 2.0, this is readily feasible (Fig. 3d). However, when the skew factor is larger than 2.0, artifacts may manifest due to the interpolation of voxels that are spatially distant within the actual sample space during the combined operations, as depicted in Extended Data Fig. 2a,b. Thus, in this case, we first interpolate the raw skewed data between adjacent planes within the proper coordinate system to add additional planes to reduce potential artifacts in the following combined operations (Fig. 3d).

Our combined deskew and rotation method yielded nearly identical results to the same operations performed sequentially (Fig. 3e and Extended Data Fig. 2c–e). The combined operation is an order of magnitude faster and becomes increasingly more efficient in speed and memory compared to sequential operation as the number of frames increases. This enables us to process ten times larger data with the same computational resources (Fig. 3f,g and Extended Data Fig. 2f,g). By additionally combining our fast Tiff reader/writer with combined deskew/rotation, we achieve at least 20 times more gain in processing

speed compared to conventional Tiff and sequential processing, allowing us to process much larger data (Fig. 3h and Extended Data Fig. 2h). Our approach is faster than the CPU and GPU implementations in pyclesperanto[35] without succumbing to GPU memory limitations, and also outperforms the implementation in qi2lab-OPM[36] (Extended Data Fig. 2i). Finally, resampling and cropping, if necessary, can also be integrated with deskewing and rotation to optimize processing efficiency and minimize storage requirements for intermediate data.

## OTF masked RL deconvolution

Deconvolution plays a crucial role in reconstructing the most accurate possible representation of the sample from light microscopy images, especially for light sheet images with strong side lobes associated with higher axial resolution[37]. Richardson–Lucy (RL) deconvolution is the most widely used due to its accuracy and robustness[38,39]. We have found that applying RL to the raw light sheet data before combined deskew/rotation not only is faster (due to no zero padding) but also yields better results with fewer edge artifacts (Extended Data Fig. 3a). To do so, the reference point spread function (PSF) used for deconvolution must be either measured in the skewed space as well (Extended Data Fig. 3c), or calculated by skewing a PSF acquired in the sample Cartesian coordinates (Extended Data Fig. 3b).

RL is an iterative method that typically requires 10–200 iterations (Biggs accelerated version[40]) to converge, depending on the type of light sheet or image modalities. Consequently, RL deconvolution is the most computationally intensive step for large datasets relative to deskew, rotation and stitching, even with GPU acceleration. Despite being notably faster than the traditional

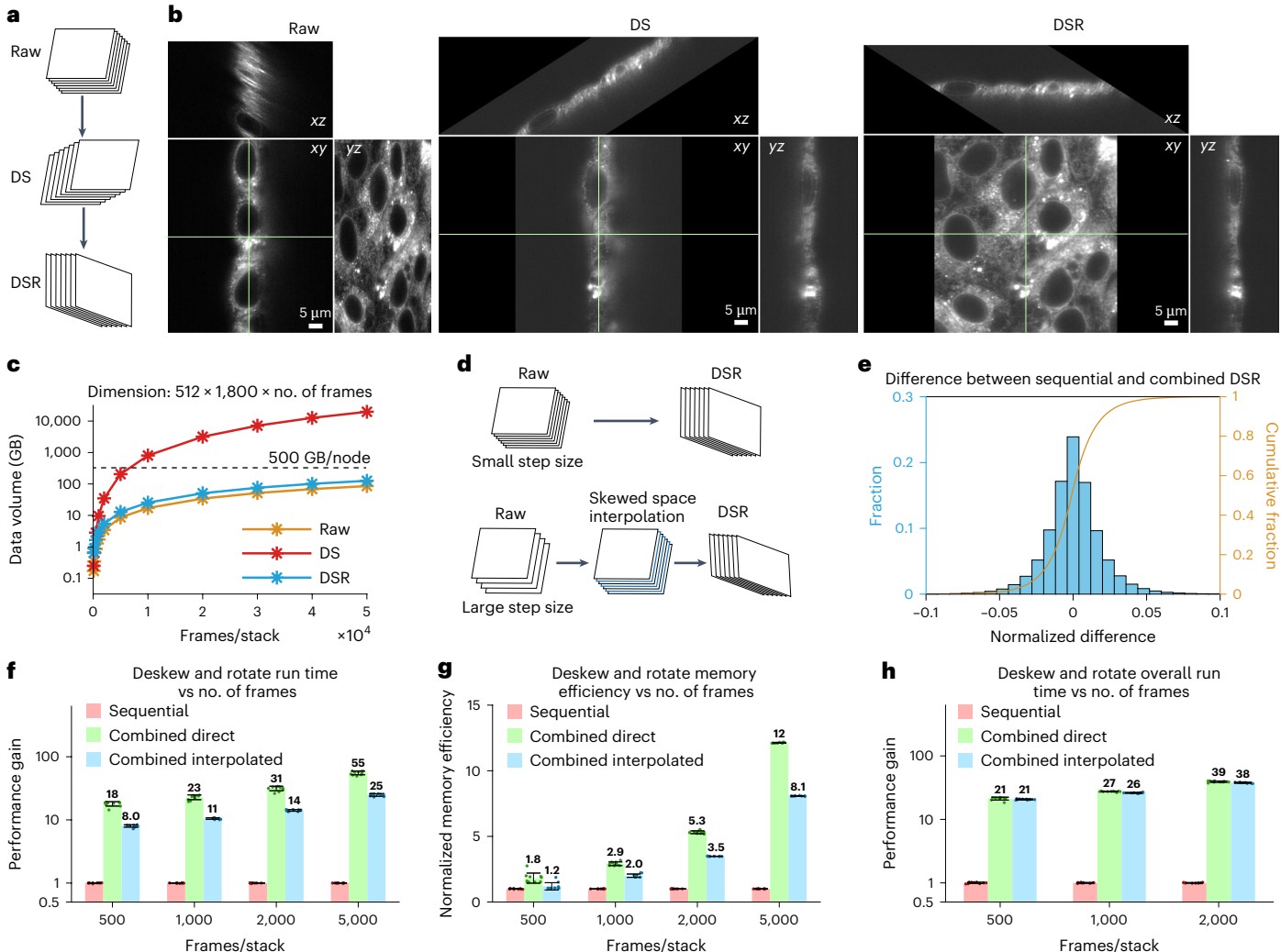

**Fig. 3 | Combined deskew and rotation. a**, Traditional sequential deskew and rotation processes. **b**, Orthogonal views of raw (skewed), deskewed (DS) and deskewed/rotated (DSR) images for cultured cells. **c**, The semi-log plot of the data sizes of a stack of raw 512 × 1,800-pixel images, deskewed, deskewed/ rotated data as a function of the number of frames per stack. The dashed line indicates the memory limit (500 GB) of a computing node. **d**, Combined deskew and rotation processes without (top) and with (bottom). The addition of skewed space interpolation before deskew/rotation when the *z*-step size between image planes is too large. **e**, Histogram of the normalized differences between deskewed/rotated results of sequential versus combined methods for the same

image in **b. f**, Performance gain for combined direct and combined interpolated over sequential deskew/rotation versus the number of frames per stack, from 500 to 5,000. Comparisons do not include read/write time, which is considered in **h. g**, Memory efficiency gain for the same three scenarios. **h**, Performance gain for the same three scenarios. This comparison does include the differences in read/write time when the conventional Tiff software is used for the sequential deskew/rotation, and our Cpp-Tiff is used for combined deskew/rotation. The benchmarks were run independently ten times on a 24-core CPU computing node (dual Intel Xeon Gold 6146 CPUs), and data are shown as the mean ± s.d. in **f**–**h**.

RL method, the Wiener–Butterworth (WB) method proposed by Guo et al.[17] with unmatched backward projector was initially demonstrated on Gaussian light sheets with ellipsoid support, and failed to achieve full-resolution reconstruction of LLSM images (Fig. 4a,b and Extended Data Fig. 3d–f) since it truncates the optical transfer function (OTF) near the edges of its support (Fig. 4a,b and Extended Data Fig. 3d–f), resulting in the loss of information. This limitation is particularly pronounced for lattice light sheets capable of high axial resolution, such as the hexagonal, hexrect and multi-Bessel types[37], whose OTF supports are nearly rectangular rather than ellipsoidal in the *xz* and *yz* planes. Another concern is that the WB method suppresses high-frequency regions near the border of its back projector's ellipsoid, thereby underweighting or even eliminating high-resolution information in the deconvolved images.

To address these issues, inspired by Zeng et al.[41] and Guo et al.[17], we optimized the backward projector by using the convex hull of the

OTF support to define an apodization function. This function filters noise close to the support and eliminates all information beyond it (Fig. 4a,b Extended Data Fig. 3d–f and 4a,b), and is applied to the Wiener filter (Extended Data Fig. 4c,d). Unlike the WB method, this OTF masked Wiener (OMW) technique covers all relevant frequencies in the Fourier space (Fig. 4c,d and Extended Data Fig. 3g,h) and achieves full-resolution image reconstruction while maintaining rapid convergence speed (Fig. 4e,f and Extended Data Fig. 3i–l). By using the OTF support for apodization, the OMW method is generic for any PSF. Our specific implementation offers a tenfold speed improvement compared to the traditional RL method (Biggs version) on both CPUs and GPUs (Fig. 4g,h and Extended Data Fig. 5). A detailed comparison with other deconvolution methods is provided in Supplementary Note 4.

The performance of RL relies crucially on finding an optimum number of iterations: too few yields fuzzy images and, in the case

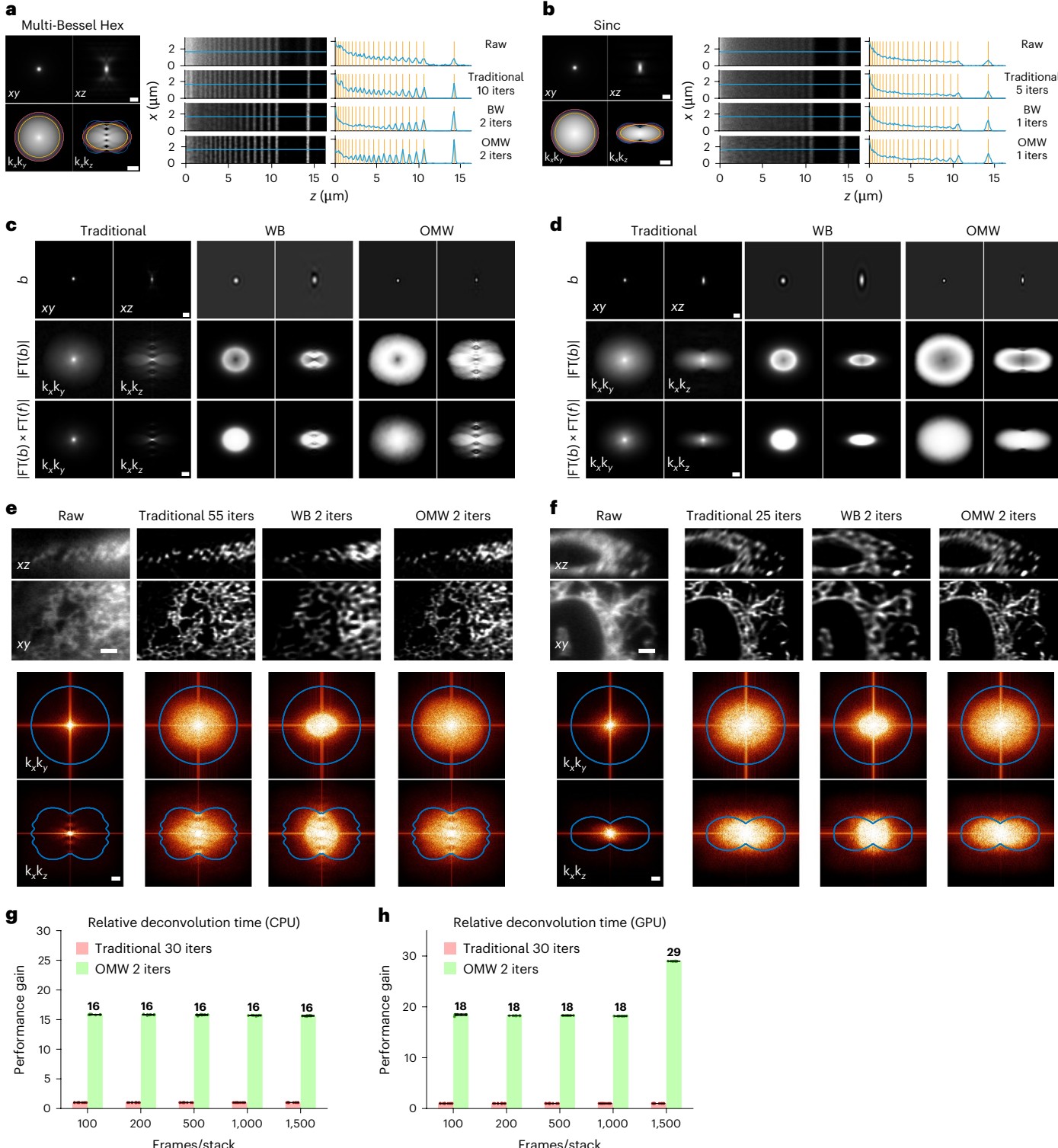

**Fig. 4 | Fast RL deconvolution. a**, Left, theoretical *xy* and *xz* PSFs (top, intensity $\gamma = 0.5$; scale bar, 1 μm) and OTFs (bottom, log-scale; scale bar, 2 μm⁻¹) for the multi-Bessel (MB) light sheet with excitation NA 0.43 and annulus NA 0.40–0.47. Blue represents theoretical support; orange and yellow indicate theoretical maximum (orange) and experimental (yellow) envelopes for the WB method; magenta indicates experimental envelope for the OMW method. Right, illustration of deconvolution of a simulated stripe pattern. The raw and deconvolved images with traditional, WB and OMW methods are displayed along with their line cuts. The orange lines indicate the theoretical line locations, and the blue curves give the actual intensities along the line cuts. **b**, Similar results for a Sinc light sheet (NA 0.32, $\sigma_{NA} = 5.0$). **c**,**d**, illustration of backward projectors (top;

scale bar, 1 μm), their Fourier spectra (middle; intensity $\gamma = 0.5$), and the products with forward projectors in Fourier spaces (bottom; intensity $\gamma = 0.5$; scale bar, 1 μm⁻¹) for the MB light sheet (**c**) and Sinc light sheet (**d**). **e**,**f**, Orthogonal views of cell images for raw, traditional RL, WB and OMW methods for the MB light sheet (**e**) and Sinc light sheet (**f**), with iteration numbers as shown (scale bar, 2 μm). The Fourier spectra are shown below each deconvolved image (intensity $\gamma = 0.5$; scale bar, 1 μm⁻¹). **g**,**h**, Relative deconvolution acceleration for traditional RL and OMW methods on CPU (**g**) and GPU (**h**) (only the deconvolution performance is shown in the comparison). Each test in **g** and **h** was run independently ten times; data are the mean ± s.d.

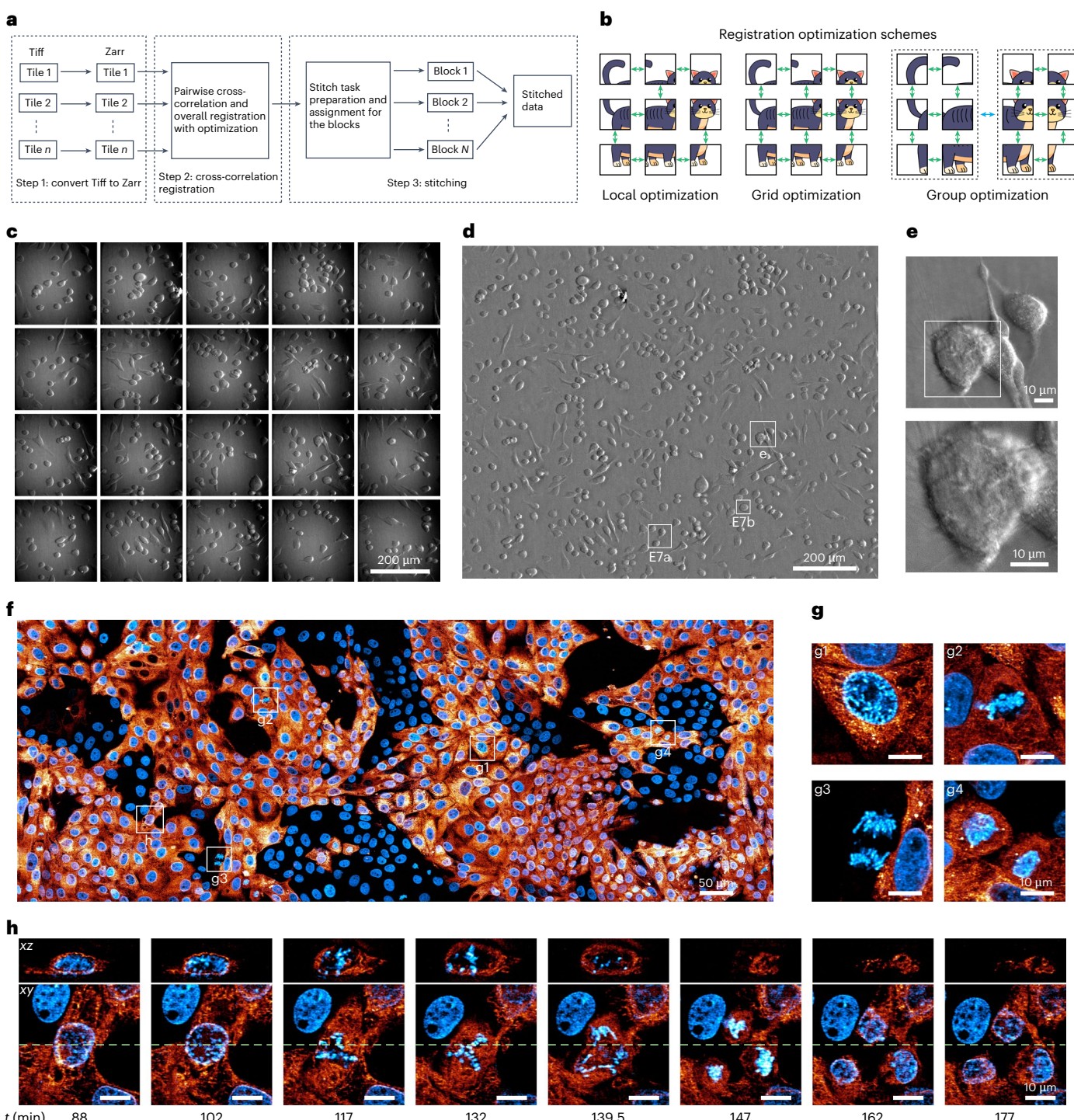

**Fig. 5 | The Zarr-based distributed stitching framework. a**, Schematic of the stitching steps. **b**, Schematic of different registration methods: local, grid and grouped. Green or blue double arrow lines indicate pairs of tiles/groups involved in registration. **c**, Raw 2D oblique illumination 'phase' tiles of live HeLa cells before processing. **d**, Final processed phase image after flat-field correction, stitching and deconvolution. Boxes labeled E7a and E7b indicate regions shown at higher magnification in Extended Data Fig. 7a,b. **e**, Zoomed-in region

of **d** showing retraction fibers. **f**, *xy* MIP view of long-term large field-of-view 3D imaging of cultured LLC-PK1 cells. Blue denotes H2B-Cherry; orange denotes Connexin-Emerald. Intensity *γ* = 0.5. Boxes labeled g1–g4 and h are the cropped and zoomed regions shown in **g** and **h**. **g**, Cropped regions from **f** showing stages in cell division. **h**, Time-lapse orthogonal views of one cell division into three daughter cells from **f**. The green dashed lines indicate the orthogonal slice positions in the *xz* plane.

of LLSM, incomplete sidelobe collapse; too many amplifies noise and potentially collapses and fragments continuous structures as represented in the deconvolved image. To find an optimum, we use Fourier shell correlation (FSC)[42]. While the traditional RL method often needs tens of iterations to optimize resolution by the FSC metric

(Supplementary Fig. 1a), the OMW method typically only needs two iterations when we use FSC to determine the optimal Wiener parameter (Supplementary Fig. 1b,c) for the backward projector for light sheet images. Widefield and confocal images, however, may need more iterations (Extended Data Fig. 6).

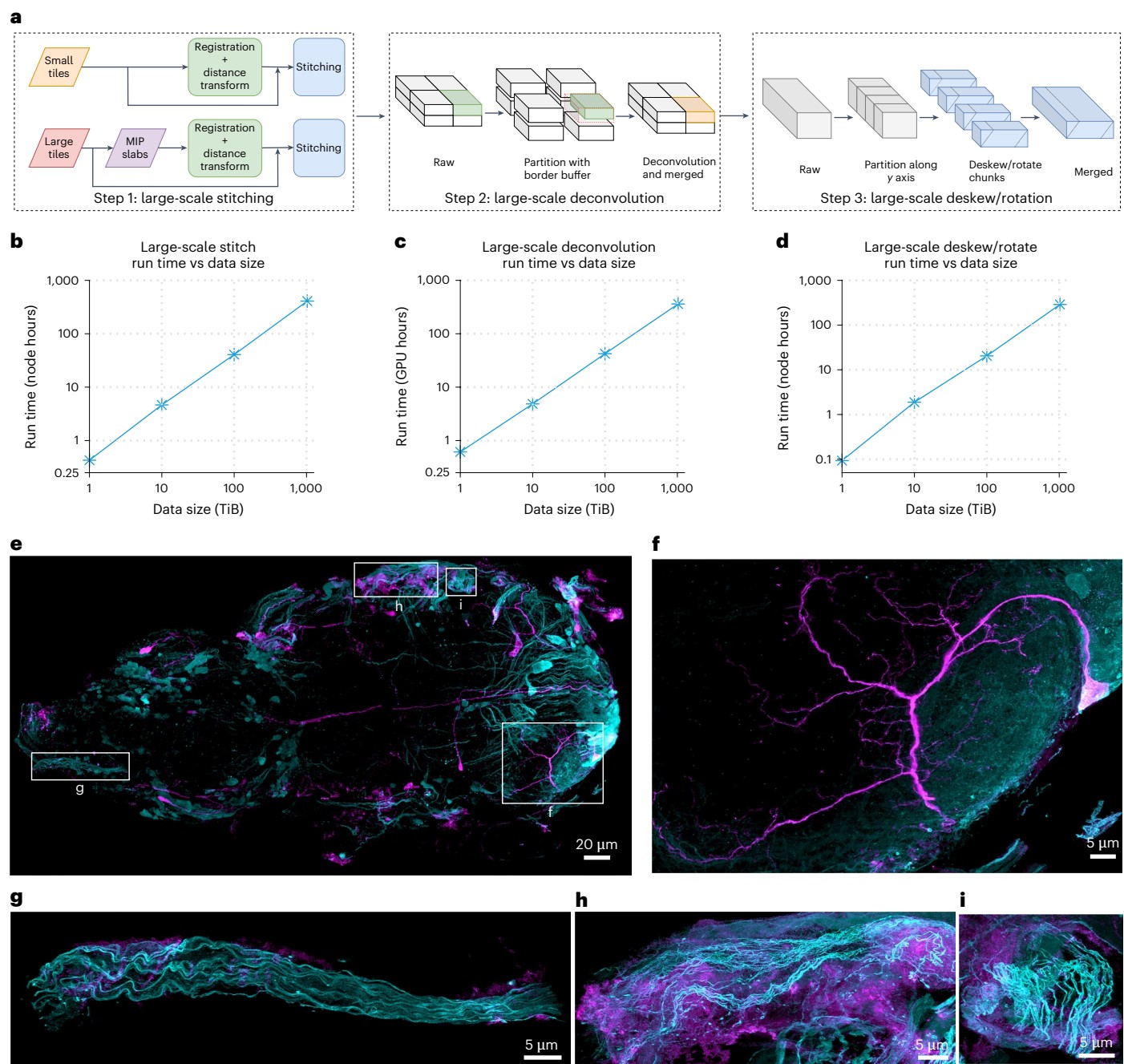

**Fig. 6 | Large-scale processing. a**, Schematic of processing steps for large-scale stitching, deconvolution and deskew/rotation. For deconvolution, the data are split into subvolumes in all three axes with an overlap border size of slightly over half of the PSF size. For deskew and rotation, the data are split along the $y$ axis with a border of one slice. **b**, Total run times for large-scale stitching of a single volume with size ranging from 1 TiB to 1 PiB. **c**, Total run times for large-scale deconvolution of a single volume with size ranging from 1 TiB to 1 PiB.

**d**, Total run times for large-scale deskew and rotation of a single volume with size ranging from 1 TiB to 1 PiB. Each benchmark in **b**–**d** was repeated three times independently. The resulting standard deviations are smaller than the data markers in the plots. **e**, MIP view of the entire fly VNC at 8× expansion. Cyan indicates VGlut[MI04979]-LexA::QFAD, and purple indicates MN-GAL4. Intensity $\gamma = 0.5$. **f**–**i**, MIP views of cropped regions from **e**. Intensity $\gamma = 0.75$ for all four regions.

## ZarrStitcher: Zarr-based scalable stitching

To image specimens such as organoids, tissues or whole organisms larger than the field of view of the microscope, it is necessary to stitch together multiple smaller image tiles. Overlapping regions between adjacent tiles facilitate precise registration and stitching. With the combination of high-resolution light sheet and expansion microscopy, thousands of tiles comprising hundreds of terabytes of data may be generated. This presents substantial challenges for existing stitching software, particularly with respect to the large number of tiles, the overall data size and the need for computational efficiency. To address these issues, we developed ZarrStitcher, a petabyte-scale framework for image stitching.

ZarrStitcher involves three primary steps (Fig. 5a): data format conversion, cross-correlation registration and stitching (fusion). We first convert tiles into the computationally efficient Zarr format, while also applying user-defined preprocessing functions such as flat-field correction and data cropping if necessary. Next, we use the normalized cross-correlation algorithm[43] to correct for sample movement

and stage motion errors and thereby accurately register the relative positions of adjacent tiles. We then apply a global optimization to infer the optimal shifts (Fig. 5b) of all tiles collectively. This better manages potential discrepancies between neighboring tiles than the typical 'greedy' local approach (Fig. 5b). For large volumes, data are often collected in multiple batches, each consisting of multiple tiles, sometimes with differing rectangular grids in each batch. In such cases, we implement two-step optimization, where global optimization is first applied to each batch, followed by optimization across batches (Fig. 5b).

The final operation in ZarrStitcher involves stitching the registered tiles together into a single unified volume. We developed a scalable distributed architecture to this end, with individual tasks allocated to different workers for different subregions. The software incorporates multiple methods to address overlapping regions, including direct merging, mean, median or feather blending. Feather blending, a type of weighted averaging with weights determined by distances to the border, has shown to be particularly effective[44].

ZarrStitcher is substantially faster than BigStitcher-Spark (Spark version of BigStitcher)[19] (Supplementary Table 2): in the case of the 108-TiB dataset for the entire mouse brain imaged with 4× expansion using ExA-SPIM ([13]), ZarrStitcher took 1.4 h using 20 computing nodes (480 CPU cores) to assemble the complete volume, 14.3 times faster than BigStitcher-Spark. This is an active research area, with ongoing development efforts working to close the performance gap (Supplementary Table 2). Stitching-spark[12], another alternative, is not usable at this scale, due to its use of Tiff files that are limited to 4 GB in size. ZarrStitcher outperforms BigStitcher-Spark in fusing images in cases with extensive overlap, minimizing ghost image artifacts caused by imperfect structure matches in overlapping regions (Supplementary Fig. 2).

By integrating fast readers and writers, combined deskew and rotation, and ZarrStitcher, we assembled a pipeline with real-time feedback during microscopy acquisition that facilitates rapid analysis and decision-making. In the online processing mode, this pipeline uses the native coordinates for stitching without global registration. It allows acquisition errors to be identified mid-stream, so that corrections can be made (Supplementary Fig. 3a) and helps determine when the specimen has been fully imaged so the acquisition can be concluded (Supplementary Fig. 3b). It also enables quick identification of specific cells or specific events in a large field of view worthy of more detailed investigation (Fig. 5c–e), such as cell fusion (Extended Data Fig. 7a) or cell division (Extended Data Fig. 7b and Supplementary Video 1).

We have also coupled our processing pipeline to NVIDIA's multi-GPU IndeX platform[45] to enable real-time visualization of 4D petabyte-scale data at full resolution (Supplementary Note 5 and Supplementary Video 3). This allows us to simultaneously follow the dynamics of hundreds to thousands of cells (Fig. 5f and Supplementary Videos 2 and 3), and identify infrequent or rare events such as normal cell divisions or the division of a cell into three daughter cells (Fig. 5g,h and Supplementary Videos 2 and 3). Furthermore, it enables us to explore their 3D high-resolution subcellular structures in detail over an extended period (Supplementary Videos 2 and 3). The entire processing and imaging pipeline is applicable to many microscope modalities in addition to light sheet microscopy. These include high-speed, large field-of-view oblique illumination 'phase' imaging (Fig. 5c–e and Supplementary Video 1), large volume adaptive optical two-photon microscopy (Extended Data Fig. 8 and Supplementary Video 4), widefield imaging (Extended Data Fig. 6a,b) and confocal imaging (Extended Data Fig. 6c,d).

### Strategies for large-scale processing

For large datasets consisting of many tiles, it is most efficient to stitch the tiles in skewed space before deconvolution (Fig. 6a), thereby eliminating duplicated effort in overlapping regions as well as potential edge artifacts. Deconvolving the stitched volume in skewed coordinates

immediately thereafter is most efficient (Extended Data Fig. 3a), because the data are more compact than after deskewing. Thus, the optimal processing sequence is stitching (if necessary), followed by deconvolution, and finally combined deskew and rotation (Fig. 6a).

When handling datasets that exceed memory capacity, certain processing steps become challenging. ZarrStitcher already enables stitching data that exceed memory limitations as long as the intermediate steps can be fitted into memory. For stitching with even larger tiles, we developed a maximum intensity projection (MIP) slab-based stitching technique (Fig. 6a) where tiles are downsampled by different factors for different axes (for example, 2× for the $xy$ axes and 100× for $z$) to generate MIP slabs that fit into memory. These slabs are used to calculate registration information and estimate distance-based weights for feather blending, ensuring accurate stitching of the complete dataset (Supplementary Fig. 2b–f).

For deskew, rotation and deconvolution, we distributed subvolumes of large data among multiple workers for faster processing and merged the results into the final output (Fig. 6a). Zarr seamlessly enables this process.

In many imaging scenarios, a substantial amount of data beyond the boundary of the specimen is empty to ensure complete coverage. Processing these empty regions is unnecessarily inefficient, particularly for deconvolution (Supplementary Fig. 4a,b,f). We, therefore, define the boundary based on MIPs across all three axes and skip the empty regions for large-scale deskew/rotation (Supplementary Fig. 4c–e) and deconvolution (Supplementary Fig. 4f–h).

With the above techniques, petabyte-scale processing becomes feasible and efficient. Processing time scales linearly for stitching, deconvolution and deskew/rotation for data sizes ranging from 1 TiB to 1 PiB (Fig. 6b–d).

As an example, we processed a 38-TiB image volume of the *Drosophila* adult ventral nerve cord (VNC) at 8× expansion (Fig. 6e–i and Supplementary Video 5). All glutamatergic neurons, which include all motor neurons, are shown in cyan, and a subset of VNC neurons that include a small number of these motor neurons is shown in purple. The ability to image, process and visualize major complete anatomical regions such as the VNC at nanoscale resolution in multiple colors at such speeds opens the door to study the stereotypy and variability of neural circuits at high resolution over long distances, across large populations, different sexes and multiple species.

## Discussion

PetaKit5D achieves real-time processing at the multi-terabyte-per-hour acquisition rates of modern scientific cameras, for the extended times and/or large volumes that produce petabyte-scale datasets. It can be applied to many imaging modalities but includes deskew and rotation operations specifically useful in light sheet microscopy.

One limitation of the current pipeline is that it only supports rigid registration to compensate for sample translation, which performs well in most scenarios. However, it may not be suitable for multi-view registration or fusion, image tiles with rotation, shrinking, swelling or warping, which would require nonrigid methods such as elastic registration[46]. We anticipate addressing these limitations in future versions by developing nonlinear registration and multi-view fusion functionalities for petabyte-scale datasets. While zstd compression in Zarr is helpful, storing raw and intermediate data for petabyte-scale or larger datasets may still require hundreds of terabytes to petabytes of storage. Real-time preprocessing of raw data followed by massive compression during acquisition may be necessary to tackle this challenge.

Notably, our software is at least tenfold more efficient computationally than existing processing solutions, which can be used to either increase experimental throughput or decrease the number (and hence the cost) of computing nodes needed. In the former case, high throughput could prove useful in obtaining high-quality training data for deep learning image processing tasks[47–49], such as deconvolution[50],

denoising[51] or registration[52]. In future releases, we aim to support a broader range of capabilities to extract biologically meaningful insights from petabyte-scale 4D and 5D datasets, including segmentation, classification, tracking and image restoration by leveraging machine learning models. The speed of PetaKit5D is also attractive for combining with multi-GPU 4D visualization[45] to monitor vast image-based biological experiments in real time, including high-throughput, high-resolution 3D drug screening[53], large tissue or whole organism spatial transcriptomics[54] or long-term imaging of subcellular dynamics in live multicellular organisms[55,56].

## Online content

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

## Methods

### Generic computing framework

Our generic computing framework supports both single machines and large-scale Slurm-based computing clusters with CPU and/or GPU node configurations. The conductor job orchestrates the processing after it receives a collection of function strings (a MATLAB function call or a Bash command executed by each worker), input file names, output file names and relevant parameters for job settings, such as required memory, the number of CPU cores and the system environment. The conductor job initially checks for the presence of output files, skipping those that already exist. In single-machine setups or when Slurm job submission is disabled, the conductor job will sequentially execute tasks. Conversely, in cluster environments with the Slurm job scheduler, the conductor job formats and submits Slurm commands based on the function strings and job parameters, delegating tasks to workers in the cluster. It continuously monitors these jobs, ensuring the completion of the tasks. If a worker job fails, the conductor job resubmits it with an increased memory and CPU resources, often doubling the original specifications, until all tasks are completed, or a preset maximum retry limit is reached. Additionally, the framework allows users to define a custom configuration file. This feature tailors Slurm-related parameters to specific needs, ensuring adaptability to various user-defined function strings and compatibility with different Slurm-based computing clusters.

### Fast Tiff and Zarr readers and writers

Our Tiff reader/writer leverages the capabilities of the libtiff library in C++ with the MATLAB MEX interface. When reading, a binary search is used to determine the number of $z$-slices by identifying the last valid slice, as there is no direct way to query the number of $z$-slices in libtiff. The OpenMP framework is then used to distribute the reading tasks across multiple threads, partitioning the $z$-slices into evenly sized batches (except for the last one). For large 2D images, the Tiff strips are partitioned to facilitate multi-threaded reading using the OpenMP framework. For the Tiff writer, LZW compression from libtiff is adapted to support compression on individual $z$-slices. This approach enables parallel compression across $z$-slices, leveraging the OpenMP framework for multi-threading. The final compressed data are written to disk using a single thread because a Tiff file is a single container, making parallel writing of compressed data infeasible.

As MATLAB lacks a native Zarr reader and writer, we developed custom C++ code that complies with the Zarr specification (version 2) with enhanced parallelization. This code is also integrated with MATLAB through the MEX interface. In our implementation, the OpenMP framework is used for both reading and writing to distribute the tasks across multiple threads, treating each chunk as a separate task. We use the compression algorithms from the Blosc library[57], which introduces an additional layer of multi-threading, thus optimizing the use of system resources. Zstd compression with a level 1 setting is used to achieve an optimal balance of compression ratio and read/write time. The high compression ratio of zstd substantially reduces the overall data size, reducing network load, particularly in extensive high-throughput processing scenarios where the network is often the primary bottleneck. By default, we read and write Zarr files in the 'Fortran' (column-major layout) order because MATLAB is based on 'Fortran' order, and converting between 'C' (row-major layout) and 'Fortran' orders adds additional overhead.

### Combined deskew, rotation, resampling and cropping

We execute deskew, rotate and resampling (if needed) in a single step by combining these geometric transformations. The fundamental geometric transform involves:

$$I_t = F_T(I)$$

where $I$ is the original image, $I_t$ represents the transformed image, $F_T(\cdot)$ denotes the image warp function corresponding to the geometric transformation matrix $T$. The deskew operator applies a shear transformation defined by the shear transformation matrix $S_{ds}$. In the rotation process, there are four sub-steps: translating the origin to the image center, resampling in the $z$ axis to achieve isotropic voxels, rotating along the $y$ axis, and translating the origin back to the starting index. Let the transformation matrices be denoted as $T_1$, $S$, $R$ and $T_2$, respectively. If resampling factors are provided (by default as 1), then there are three additional sub-steps in resampling: translating the origin to the image center, resampling based on the factors provided, and translating the origin back to the start index. Let the transformation matrices in these sub-steps be $T_{R1}$, $S_R$ and $T_{R2}$, respectively.

Traditionally, these three steps are executed independently, resulting in multiple geometric transformations. However, this incurs substantial overhead in run time and memory usage, particularly during the deskew step. Instead, we combine deskew, rotation and resampling into one single step, resulting in a unified affine transformation matrix:

$$A = S_{ds}(T_1 S R T_2)(T_{R1} S_R T_{R2})$$

This affine transformation matrix can be directly applied to the raw image if the scan step size is sufficiently small. A quantity, denoted as 'skew factor', is defined to describe the relative step size as

$$f_{sk} = d_z \cos\theta / p_x$$

where $\theta \in (-\pi/2, \pi/2]$ is the skewed angle, $d_z$ denotes the scan step size, and $p_x$ is the pixel size in the $xy$ plane. If $f_{sk} \leq 2$, the direct combined processing operates smoothly without noticeable artifacts. For $f_{sk} > 2$, interpolation of the raw data within the skewed space is performed before deskew and rotation, taking account of the proper relative positions of slices. Neighboring slices above and below are utilized to interpolate a $z$-slice. Let $w_s$ and $w_t = 1 - w_s$ represent the normalized distances (ranging from 0 to 1) along the $z$ axis between the neighboring slices and the target $z$-slice. In the interpolation, we first create two planes aligned with the correct voxel positions of the target $z$-slice by displacing the neighboring slices with a specific distance in the $x$ direction ($w_s d_z \cos\theta$ and $w_t d_z \cos\theta$, respectively). Following this, the target $z$-slice is obtained by linearly interpolating these two planes along the $z$ axis with weights $1 - w_s$ and $1 - w_t$. Because the image warp function permits the specification of the output view, we have also incorporated a cropping feature by providing a bounding box that allows us to skip empty regions or capture specific regions.

For combined deskew and rotation without resampling, the transformation simplifies to a 2D operation in the $xz$ plane. We optimized this scenario using SIMD (single instruction, multiple data) programming in C++. Our implementation directly supports uint16 input and output for both skewed space interpolation and deskew/rotation functions, while utilizing single-precision floating-point for intermediate steps. This improves throughput and reduces memory usage.

In acquisition modes where the deskew operation is unnecessary (for example, objective scan mode of LLSM), the above processing can still be applied, provided $S_{ds}$ is replaced with the identity matrix.

### Deconvolution

RL deconvolution has the form of where $I$ is the raw data, $f$ is the forward projector (that is, the PSF), $b$ is the backward projector and $b^T$ is the transpose of $b$, $\circledast$ denotes the convolution operator and $x^{(k)}$ is the deconvolution result in $k$-th iteration. In traditional RL deconvolution, $b = f$. In the OMW method we use, the backward projector is generated with these steps:

1. The OTF $H$ of the PSF $f$ is computed, $H = \mathcal{F}(f)$, where $\mathcal{F}(\cdot)$ represents the Fourier transform.
2. The OTF mask for the OTF support is segmented by applying a threshold to the amplitude $|H|$. The threshold value is determined by a specified percentile (90% by default) of the accumulated sum of sorted values in $|H|$ from high to low.

3. The OTF mask undergoes a smoothing process, retaining only the central object, followed by convex hull filling. For deskewed space deconvolution, the three major components are kept after object smoothing and concatenated into a unified object along the $z$ axis, followed by convex hull filling.

4. The distance matrix $D$ is computed with the image center as 0, and the edge of the support as 1 with the ray distance from the center to the border of the whole image.

5. The distance matrix $D$ is used to calculate the weight matrix $W$ with the Hann window function for apodization, as expressed by the following formula:

$$w(x) = \begin{cases} 1 & x \le l \\ cos^2\left(\frac{\pi(x-l)}{2(u-l)}\right) & l < x \le u \\ 0 & x > u \end{cases}$$

Where $l$ and $u$ are the lower and upper bounds for the relative distances. By default, $l = 0.8$ and $u = 1$ (edge of the support). For skewed space deconvolution, the weight matrix is given as a single distance matrix by adding the distance matrix from the corresponding three components together.

6. Calculate Wiener filter $F = \frac{H^*}{|H|^2 + \alpha}$, where $\alpha$ is the Wiener parameter, and $H^*$ denotes the conjugate transpose of $H$.

7. The backward projection in the Fourier space is expressed as $B = W \odot F$, where $\odot$ denotes the Hadamard product operator (element-wise multiplication), and the backward projector in the real space is $b = \mathscr{F}^{-1}(B)$, where $\mathscr{F}^{-1}(\cdot)$ represents the inverse Fourier transform.

The FSC method[42] is used to determine the optimal number of traditional RL iterations and the optimal Wiener parameter in the OMW method. Here, the central portion of the volume, which is consistent in size across all three axes and covers sufficient content (for example, 202 × 202 × 202 for a volume with size 230 × 210 × 202), is used to compute the relative resolution. By default, the FSC is calculated with a radius of ten pixels and an angle interval of $\pi/12$. Cutoff frequencies for relative resolution are determined using one-bit thresholding[58] by default, or can be user-defined. The relative resolution across iterations (or different Wiener parameters) is plotted. In ref. 37, it was determined that a slightly higher threshold produced better results (Supplementary Fig. 1a). In practice, the optimal number of RL iterations or the Wiener parameter is defined by the value closest to 1.01 times the minimum of the curve beyond the point where the curve reaches its minimum value.

## Stitching

The stitching process requires a CSV meta-file documenting file names and corresponding coordinates. The pipeline consists of three steps: Tiff to Zarr conversion (or preprocessing), cross-correlation registration, and parallel block stitching (fusion). The overall stitching workflow is governed by a conductor job in the generic computing framework. For Tiff to Zarr conversion and/or processing on individual tiles, the conductor job distributes tasks to individual worker jobs, assigning one worker for each tile. Each worker: (a) reads its data using the Cpp-Tiff or Cpp-Zarr (if existing Zarr data need rechunking or preprocessing) reader depending on the format; (b) performs optional processing such as flipping, cropping, flat-field correction, edge erosion or other user-defined operations; and (c) writes the processed data using the Cpp-Zarr writer.

Following file conversion, stitching can be executed directly using the input tile coordinates, or normalized cross-correlation registration[43] can be used first to refine and optimize the coordinates before stitching. In the registration, the conductor job utilizes coordinate information and tile indices to establish tile grids and identify neighboring tiles with overlaps. Cross-correlation registration is performed for overlapping tiles that are direct neighbors, defined as those whose tile indices differ by 1, and only in one axis. To optimize computing time and memory usage, only the overlapping regions for the tiles are loaded, including a buffer size determined by the maximum allowed shifts along the $xyz$ axes within one tile. We can also downsample the overlapping data to achieve faster cross-correlation computing. The optimal shift between the two tiles is identified as the one exhibiting the maximum correlation within the allowable shift limits. We include a feature to exclude shifts for pairs with the maximum correlation values below a user-defined threshold. After completing the cross-correlation computation for all pairs of direct neighbor tiles, we determine the shifts for all tiles using either a local or a global method. The local approach is based on the concept of the minimum spanning tree, where the pairs of overlapping tiles are pruned to form a tree based on the correlation values from high to low, followed by registration with the pairwise optimal shifts. In the global approach, the optimal final shifts are calculated from the pairwise relative shifts through a nonlinear constrained optimization process:

$$\min_x \sum_{i,j} w_{ij} \parallel x_i - x_j - d_{ij} \parallel_2^2$$

$$s.t. \quad l < x_i - x_j < u$$

where $\mathbf{x}_{shift} = \{x_1, \dots, x_n\}$ are the final shifts for the tiles, $d_{ij}$ is the pairwise relative shift between tile $i$ and $j$ and $w_{ij}$ is the weight between tile $i$ and $j$ based on the squares of maximum cross-correlation values. $l$ and $u$ are the lower and upper bounds for the maximum allowable shift, respectively. The goal is to position all tiles at optimal coordinates by minimizing the weighted sum of the squared differences between their distances and the pairwise relative shifts while adhering to the specified maximum allowable shifts.

For images collected by subregions (batches) that have different tile grids, we use the global method for tiles within each subregion. Subsequently, the subregions are treated as super nodes, and a nonlinear constrained optimization is applied to those nodes, by minimizing the sum of squared differences of the centroid distances to the averaged shift distances.

$$\min_x \sum_{i,j} \parallel x_{ri} - x_{rj} - d_{r,ij} \parallel_2^2$$

$$s.t. \quad l_r < x_{ri} - x_{rj} < u_r$$

where $x_{ri}$ and $x_{rj}$ are the centroid coordinates for subregions $i$ and $j$, and $l_r$ and $u_r$ are lower and upper bounds for the maximum allowable shifts across subregions, respectively. The averaged shift distance, denoted as $d_{r,ij}$, is determined by a weighted average of the absolute shifts across subregions, which is expressed as:

$$d_{r,ij} = \frac{\sum_{m \in S_i, n \in S_j} w_{mn} d_{mn}}{\sum_{m \in S_i, n \in S_j} w_{mn}}$$

where $w_{mn}$ is the cross-correlation value at the optimal shift between tiles $m \in S_i$, and $n \in S_j$, and $S_k$ denotes the set of tiles in subregion $k$. Once the optimal shifts for the subregions are obtained, the last step is to reconstruct the optimal shifts for the tiles within each subregion by applying the optimal shifts of the centroid of the subregion to the coordinates of the tiles in it. The final optimal shifts are then applied to the tile coordinates to determine their final positions.

After registration, the conductor job determines the final stitched image size and the specific locations to place the tiles. To facilitate parallel stitching, the process is executed region by region in a nonoverlapping manner. These regions are saved directly as one or more

distinct chunk files in Zarr format. For each region, information about the tiles therein and their corresponding bounding boxes are stored. The conductor job submits stitching tasks to worker jobs. If the region comes from one tile, the data for the region are saved directly. If the region spans multiple tiles, these must be merged into a single cohesive region. For the overlap regions, several blending options are available: 'none', 'mean', 'median', 'max' and 'feather'. For the 'none' option, half of the overlap region is taken from each tile. For the 'mean', 'median' and 'max' options, the voxel values in the stitched region are calculated as the mean, median and maximum values from the corresponding voxels in the overlapping regions, respectively. Feather blending involves calculating the weighted average across the tiles[44]. The weights are the power of the distance transform of the tiles as follows:

$$w_{i,m} = d_{i,m}^{\alpha}/\left(d_{i,m}^{\alpha} + d_{j,n}^{\alpha}\right) \quad \text{and} \quad w_{j,n} = d_{j,n}^{\alpha}/\left(d_{i,m}^{\alpha} + d_{j,n}^{\alpha}\right)$$

$$I_{s,l} = w_{i,m}I_{i,m} + w_{j,n}I_{j,n}$$

where $d_{i,m}$ and $d_{j,n}$ are distance transforms for voxel $m$ in tile $i$ and voxel $n$ in tile $j$, $\alpha$ is the order (10 by default), $I_i$ and $I_j$ are the intensities for tiles $i$ and $j$, and $I_s$ is the intensity for the stitched image $s$. Here we assume voxel $m$ in tile $i$ and voxel $n$ in tile $j$ are fused to voxel $l$ in the stitched image. For the distance transform, we utilize a weighted approach, applying the distance transform to each $z$-slice and then applying the Tukey window function across $z$-slices to address the anisotropic properties of voxel sizes. When all tiles are the same size, we compute the weight matrix for a single tile and apply it across all other tiles in the stitching process to save computing time. The final stitched image is obtained once all the subvolumes are processed.

### Large-scale processing
For stitching involving large tiles where intermediate steps above exceed memory capacities, including large, stitched subregions, challenges arise in the registration and calculation of the distance transformation for feather blending, due to the need to load large regions or tiles into memory. In such cases, we use MIP slabs for the registration and distance transform. These are computed across all three axes with downsampling factors $[M_x, M_y, M_z, m_x, m_y, m_z]$. The MIP slab for each specific axis is computed using the major downsampling factor $M_i$ for that axis, and the minor downsampling factors $m_j$ and $m_k$ for the other two. To enrich the signal for cross-correlation in sparse specimens, we use maximum pooling, that is, taking the max value in the neighborhood for the downsampling. Alternatively, we can also smooth the initial data by linear interpolation before maximum pooling. For the registration, normalized cross-correlation is calculated between direct neighbor tiles using all three MIP slabs, generating three sets of optimal shifts. The optimal shifts from the minor axes are then averaged to obtain the final optimal shifts, with weights assigned based on the squares of the cross-correlation values. For the distance transform, only the MIP slab along the $z$ axis (major axis) is used to compute the weights for feather blending. In the stitching process, for overlapping regions, the downsampled weight regions are upsampled using linear interpolation to match the size of the regions in the stitching. The upsampled weights are then utilized for feather blending, following the same approach as that used for stitching with smaller tiles.

For large-scale deskew and rotation, tasks are divided across the $y$ axis based on the size in the $x$ and $z$ axes, with a buffer of one or two pixels on both sides in the $y$ axis. These tasks are then allocated to individual worker jobs for processing, with the results saved as independent Zarr regions on disk. MIP masks can be used to define a tight boundary for the object to optimize efficiency in data reading, processing and writing. We also perform deskewing and rotation for the MIP along the $y$ axis to define the bounding box for the output in the $xz$ axes. The geometric transformation function directly relies on this

bounding box to determine the output view to minimize the empty regions, thereby further optimizing processing time, memory and storage requirements.

For large-scale deconvolution, tasks are distributed across all three axes, ensuring that regions occupy entire chunk files. An additional buffer size, set to at least half of the PSF size (plus some extra size, 10 by default), is included to eliminate edge artifacts. MIP masks are again used to define a tight specimen boundary to speed computing. In a given task, all three MIP masks for the region are loaded and checked for empty ones. If a mask is empty, deconvolution is skipped, resulting in an output of zeros for that region.

### Image processing and simulations
All images were processed using PetaKit5D. Flat-field correction was applied for the large field-of-view cell data (Fig. 5f–h), phase contrast data (Fig. 5c–e) and VNC data (Fig. 6e–j) with either experimentally collected flat-fields or ones estimated based on the data using BaSiC software[59].

The images used to benchmark different readers and writers, deskew/rotation and deconvolution algorithms were generated by cropping or replicating frames from a uint16 image of size $512 \times 1,800 \times 3,400$. The stripped line patterns used to compare deconvolution methods were simulated using the methodology outlined in ref. 37. The confocal PSF for the given pinhole size used in the stripped line pattern simulation was generated based on the theoretical widefield PSF. We benchmarked large-scale stitching from 1 TiB to 1 PiB using one channel of the VNC dataset with 1,071 tiles, each sized at $320 \times 1,800 \times 17,001$. The datasets were created by either including specific numbers of tiles or replicating tiles across all three axes based on the total data size from 1 TiB to 1 PiB, as specified in Supplementary Table 4. We benchmarked large-scale deconvolution and deskew/rotation using the stitched VNC dataset ($15,612 \times 28,478 \times 21,299$, uint16) by either cropping or replicating the data in all three axes to generate the input datasets, as indicated in Supplementary Table 4.

### Computing infrastructures
Our computing cluster has 38 CPU/GPU computing nodes: 30 CPU nodes (24 nodes with dual Intel Xeon Gold 6146 CPUs, 6 nodes with dual Intel Xeon Gold 6342 CPUs) and 8 GPU nodes (3 nodes with dual Intel Xeon Silver 4210R and 4 NVIDIA Titan V GPUs each, 4 nodes with dual Intel Xeon Gold 6144 and 4 NVIDIA A100 GPUs each, and 1 NVIDIA DGX A100 with dual AMD EPYC 7742 CPUs and 8 NVIDIA A100 GPUs). The Intel Xeon Gold 6146 CPU and GPU nodes have 512 GB RAM on each node, the Intel Xeon Gold 6342 CPU nodes have 1,024 GB RAM on each node, and the NVIDIA DGX A100 has 2 TB RAM. The hyperthreading on all Intel CPUs was disabled. Benchmarks were performed on hardware aged approximately 3 to 4 years. We have four flash data servers, including a 70 TB (SSD, Supermicro), two 300 TB (NVMe, Supermicro) and a 1,000 TB parallel file system (VAST Data). We also accessed the Perlmutter supercomputer from the National Energy Research Scientific Computing Center (NERSC), with both CPU and GPU nodes. Each CPU node is equipped with two AMD EPYC 7713 CPUs and 512 GB RAM; each GPU node has a single AMD EPYC 7713 CPU, four NVIDIA A100 GPUs and 256 or 512 GB RAM.

### Microscope hardware
Light sheet imaging was performed on a lattice light sheet microscope comparable to a published system[55]. Two lasers, 488 nm and 560 nm (500 mW, MPB Communications 2RU-VFL-P-500-488-B1R, and 2RU-VFL-P-1000-560-B1R), were used as the light sources. Water immersion excitation (EO, Thorlabs TL20X-MPL) and detection objectives (DO, Zeiss, ×20, 1.0 NA, 1.8 mm FWD, 421452-9800-000) were used for imaging with a sCMOS camera (Hamamatsu ORCA Fusion). The oblique illumination microscopy was also performed on the modified lattice light sheet microscope using a 642-nm laser

illuminated through the EO, and imaged using an inverted DO (Zeiss, ×20, 1.0 NA, 1.8 mm FWD, 421452-9880-000). Widefield and confocal imaging were performed on an Andor BC43 Benchtop Confocal Microscope (Oxford Instruments) with a Nikon Plan Apo ×40, 1.25 NA SIL Silicone objective (Nikon, MRD73400), a 488-nm laser (Oxford Instruments, Andor Borealis) and a modified Andor Zyla sCMOS camera (Oxford Instruments, 4.1 MP, 6.5-µm pixel size). Two-photon microscopy was performed on a custom-built microscope equipped with an upright DO (Zeiss, ×20, 1.0 NA, 1.8-mm FWD, 421452-9880-000), pulsed laser (Coherent, Chameleon LS), deformable mirror (ALPAO, DM69) and MPPC modules (Hamamatsu, C13366-3050GA and C14455-3050GA). The imaging conditions for the datasets can be found in Supplementary Table 5.

### Cell culture and imaging
Pig kidney epithelial cells (LLC-PK1, a gift from M. Davidson at Florida State University) cells and HeLa cells were cultured in DMEM with Gluta-MAX (Gibco, 10566016) supplemented with 10% FBS (Seradigm) in an incubator with 5% $CO_2$ at 37 °C and 100% humidity. LLC-PK1 cells stably expressing the endoplasmic reticulum marker mEmerald-Calnexin and the chromosome marker mCherry-H2B were grown on coverslips (Thorlabs, CG15XH) coated with 200-nm diameter fluorescent beads (Invitrogen FluoSpheres Carboxylate-Modified Microspheres, 505/515 nm, F8811). When cells reached 30–80% confluency, they were imaged at 37 °C in Leibovitz's L-15 Medium without Phenol Red (Gibco catalog, 21-083-027), with 5% FBS (ATCC SCRR-30- 2020) and an antibiotic cocktail consisting of 0.1% ampicillin (Thermo Fisher, 611770250), 0.1% kanamycin (Thermo Fisher, 11815024) and 0.1% penicillin–streptomycin (Thermo Fisher, 15070063). HeLa cells were cultivated on 25-mm coverslips until approximately 50% confluency was achieved. They were imaged in the same media as above.

### Mouse brain sample preparation and imaging
All mouse experiments were conducted at Janelia Research Campus, Howard Hughes Medical Institute (HHMI) in accordance with the US National Institutes of Health Guide for the Care and Use of Laboratory Animals. Procedures and protocols were approved by the Institutional Animal Care and Use Committee of the Janelia Research Campus, HHMI. Mice were housed in a specific pathogen-free condition on individually ventilated racks with 100% outside filtered air in the holding room. They were maintained on a 12–12-h light–dark cycle at 20–22 °C with 30–70% relative humidity.

Transgenic Thy1-YFP-H mice (The Jackson Laboratory) of 8 weeks or older with cytosolic expression of yellow fluorescent protein (YFP) at high levels in motor, sensory and subsets of central nervous system neurons were anesthetized with isoflurane (1–2% by volume in oxygen) and placed on a heated blanket. An incision was made on the scalp followed by removing of the exposed skull. A cranial window made of a single 170-µm-thick coverslip was embedded in the craniotomy. The cranial window and a headbar were sealed in place with dental cement for subsequent imaging. A direct wavefront sensing method[60] was used for adaptive optical correction before image acquisition. Aberrations at each volumetric tile were independently measured and corrected using a pupil conjugated deformable mirror, and imaged at 16 Hz using Hamamatsu MPPC modules.

### Fly VNC sample preparation and imaging
A genetically modified strain of fruit flies (*Drosophila melanogaster*) was raised on a standard cornmeal-agar-based medium in a controlled environment of 25 °C on a 12–12-h light–dark cycle. On the day of eclosion, female flies were collected and group housed for 4–6 days. The genotype was VGlut[MI04979]-LexA:QFAD/MN-GAL4 (attp40); 13XLexAop-Syn21-mScarle [JK65C], 20XUAS-Syn21-GFP [attp2]/MN-GAL4 [attp2][61,62]. Dissection and immunohistochemistry of the fly VNC were performed following the protocol in ref. 63 with

minor modifications. The primary antibodies were chicken anti-GFP (1:1,000 dilution; Abcam, ab13970) and rabbit anti-dsRed (1:1,000 dilution; Takara Bio, 632496). The secondary antibodies were goat anti-chicken IgY Alexa Fluor 488 (1:500 dilution; Invitrogen, A11039) and goat anti-rabbit IgG Alexa Fluor 568 (1:500 dilution; Invitrogen, A11011). VNC samples were prepared for 8× expansion as described in ref. 63. The imaging protocol for the expanded VNC sample was identical to that described in ref. 12.

### Visualization and software
Lattice light sheet images were acquired with LabView (National Instruments) software. Videos were made with Imaris (Oxford Instruments), Fiji[33], Amira (Fisher Scientific), NVIDIA IndeX (NVIDIA) and MATLAB R2023a (MathWorks) software. Figures were made with MATLAB R2023a (MathWorks). Python (3.8.8) with Zarr-Python (2.16.1), tifffile (2023.7.10), TensorStore (0.1.45), pyclesperanto-prototype (0.24.2), qi2lab-OPM (a734490) and clij2-fft (0.26) libraries were used for benchmarking image readers and writers, deskew and rotation and deconvolution. The traditional RL deconvolution method is an accelerated version of the original RL algorithm[40,64]. It was implemented and adapted from MATLAB's 'deconvlucy.m' with enhancements such as GPU computing and customized parameters. Backward projectors for the WB deconvolution method were generated using the code from https://github.com/eguomin/regDeconProject/. Spark versions of BigStitcher (https://github.com/JaneliaSciComp/BigStitcher-Spark/) and https://github.com/saalfeldlab/stitching-spark/ were used for the stitching comparison. NVIDIA IndeX can be obtained from https://developer.nvidia.com/index/ with a free license for noncommercial research and education.

### Reporting summary
Further information on research design is available in the Nature Portfolio Reporting Summary linked to this article.

## Data availability
The full datasets for this manuscript exceed the size limits of any data repository, but they will be shared upon reasonable request. The representative subsets of the full datasets can be downloaded from https://doi.org/10.5061/dryad.kh18932g4 (time-lapse live cell imaging data, two-photon live mouse brain imaging data, oblique illumination 'phase' imaging of HeLa cells, widefield and confocal imaging data)[65] and https://doi.org/10.5061/dryad.jq2bvq8jd (VNC data)[66]. The cell data for deconvolution comparison for light sheet microscopy data are from ref. 37, and can be accessed from https://doi.org/10.6078/D1VT6K, https://doi.org/10.6078/D1MB09 and https://doi.org/10.6078/D1GM7G. The stitching comparison dataset (ExA-SPIM) is from ref. 13 and can be accessed from s3://aind-open-data/exaSPIM 615296 2022-09-28 11-47-06 using AWS CLI (https://github.com/aws/aws-cli/). AWS CLI, users can use the following command: aws s3 cp –no-sign-request s3://aind-open-data/exaSPIM_615296_2022-09-28_11-47-06/ /local/path/to/destination –recursive, or following the instructions at https://allenneuraldynamics.github.io/data.html.

## Code availability
The source code of the software is available at https://github.com/abcucberkeley/PetaKit5D. The version associated with this manuscript is available on Zenodo (https://doi.org/10.5281/zenodo.13686337)[67]. We also provide a Python version for the wrapper of the deployed version of PetaKit5D at https://github.com/abcucberkeley/PyPetaKit5D. The GUI for the software can be downloaded from https://github.com/abcucberkeley/PetaKit5D-GUI/. The Parallel Fiji Visualizer plugin can be accessed from GitHub (https://github.com/abcucberkeley/Parallel_Fiji_Visualizer/) or Zenodo (https://doi.org/10.5281/zenodo.7613228)[68]. The code for replicating the benchmark results is available on Zenodo (https://doi.org/10.5281/zenodo.13690716)[69]. The NVIDIA IndeX

software can be acquired by following the instructions in Supplementary Note 5. Example code and data for data format conversion and visualization are available on Zenodo (https://doi.org/10.5281/zenodo.12539579)[70].

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

## Acknowledgements

We thank L. He for providing access to the confocal microscope. We thank J. Lefman and the NVIDIA IndeX team for sharing the NVIDIA IndeX software. We thank J. White for managing our computing cluster. We gratefully acknowledge the support of this work by the Laboratory Directed Research and Development (LDRD) Program of Lawrence Berkeley National Laboratory under US Department of Energy contract No. DE-AC02-05CH11231. This research used resources of the National Energy Research Scientific Computing Center, a US Department of Energy Office of Science User Facility located at Lawrence Berkeley National Laboratory, operated under contract no. DE-AC02-05CH11231 using NERSC awards DDR-ERCAP0025501 and DDR-ERCAP0029442. X.R., G.L., F.G., J.L.H. and S.U. are partially funded by the Philomathia Foundation (awarded to E.B. and S.U.). X.R. and G.L. are partially funded by the Chan Zuckerberg Initiative (awarded to S.U.). X.R. and S.U. are supported by Lawrence Berkeley National Laboratory's LDRD program 7647437 and 7721359 (awarded to S.U.). M.M., T.-M.F., D.M., J.L.L. and E.B. are funded by HHMI (awarded to E.B.). F.G. is partially funded by the Feodor Lynen Research Fellowship, Humboldt Foundation. J.G.C. is funded by the California Institute for Regenerative Medicine (CIRM) Predoctoral Training Program no. EDUC4-12790. E.B. is an HHMI Investigator. S.U. is funded by the Chan Zuckerberg Initiative Imaging Scientist program 2019-198142 and 2021-244163. S.U. is a Chan Zuckerberg Biohub – San Francisco Investigator.

## Author contributions

E.B. and S.U. supervised the project. X.R. wrote the manuscript with input from all co-authors. E.B. and S.U. edited the manuscript. X.R. designed the algorithms and implemented the software. M.M. implemented the fast image readers and writers, the Parallel Fiji Visualizer plugin, the Imaris file converter and the Python wrappers under X.R.'s guidance. M.M. designed and developed the GUIs, and C.Y.A.H. contributed to the implementation. J.L.L. prepared the VNC sample and G.L. performed the imaging experiments. W.H. and A.N.K. prepared the cultured LLC-PK1 cells and F.G. performed the live cell imaging experiment. T.-M.F. performed the imaging experiments for the live mouse brain imaging and phase imaging of HeLa cells. D.M. developed the microscope software for the imaging experiments. A.K. and M.N. helped set up the workflows for the real-time visualization video using NVIDIA IndeX. J.L.H. prepared the sample and J.G.C. performed the widefield and confocal imaging experiments for the LLC-PK1 cells. X.R. performed all image processing and analysis and made the figures. X.R. and M.M. made all videos with S.U.'s input.

## Competing interests

A.K. and M.N. are employees of NVIDIA. The use of the NVIDIA IndeX software platform can be licensed free of charge for educational and noncommercial research. The scientists and engineers at NVIDIA align the development of the NVIDIA IndeX solution to the requirements in various fields of scientific visualization including those that arise from the needs of the Advanced Bioimaging Center at the University of California, Berkeley. The other authors declare no competing interests.

## Additional information

**Extended data** is available for this paper at https://doi.org/10.1038/s41592-024-02475-4.

**Correspondence and requests for materials** should be addressed to Xiongtao Ruan, Eric Betzig or Srigokul Upadhyayula.

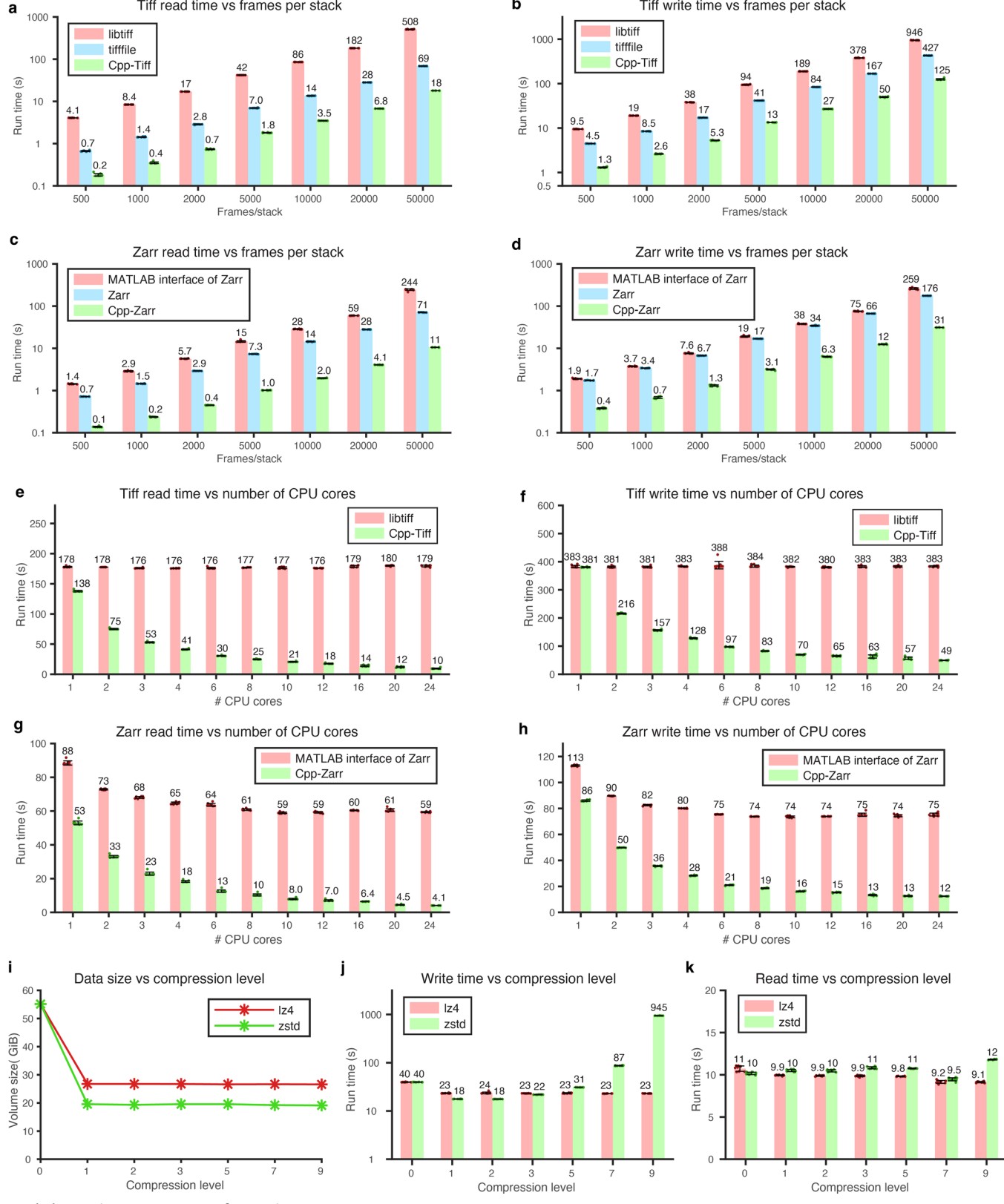

**Extended Data Fig. 1 | See next page for caption.**

**Extended Data Fig. 1 | Additional benchmarks for Tiff and Zarr readers and writers. a**–**b**, run times of Tiff readers and writers for libtiff (MATLAB), tifffile (Python), and Cpp-Tiff versus the number of frames. **c**–**d**, run times of Zarr readers and writers, comparing the MATLAB interface of Zarr, native Zarr (Python), and Cpp-Zarr across different numbers of frames. In panels **a-d**, all images are in unit16 format with a frame size of 512 × 1,800 (xy), and the benchmark results are the absolute run times for Fig. 2. **e**–**f**, run times of Tiff readers and writers for libtiff (MATLAB) and Cpp-Tiff versus the number of CPU cores for a unit16 image stack of size 512 × 1,800 × 20,000. **g**–**h**, run times

of Zarr readers and writers for the MATLAB interface of Zarr, and Cpp-Zarr versus the number of CPU cores for a unit16 image stack of size 512 × 1,800 × 20,000. **i**–**k**, data size and read/write times versus compression level for lz4 and zstd compressors for a uint16 image stack of size 512 × 1,800 × 30,000. The benchmarks were run ten times independently on a 24-core CPU computing node (dual Intel Xeon Gold 6146 CPUs). All 24 cores were allocated for panels **a**–**d** and **i**–**k**, and varying numbers of CPU cores were allocated for panels **e**–**h**. Data are shown as mean ± s.d. in panels **a**–**h** and **j**–**k**.

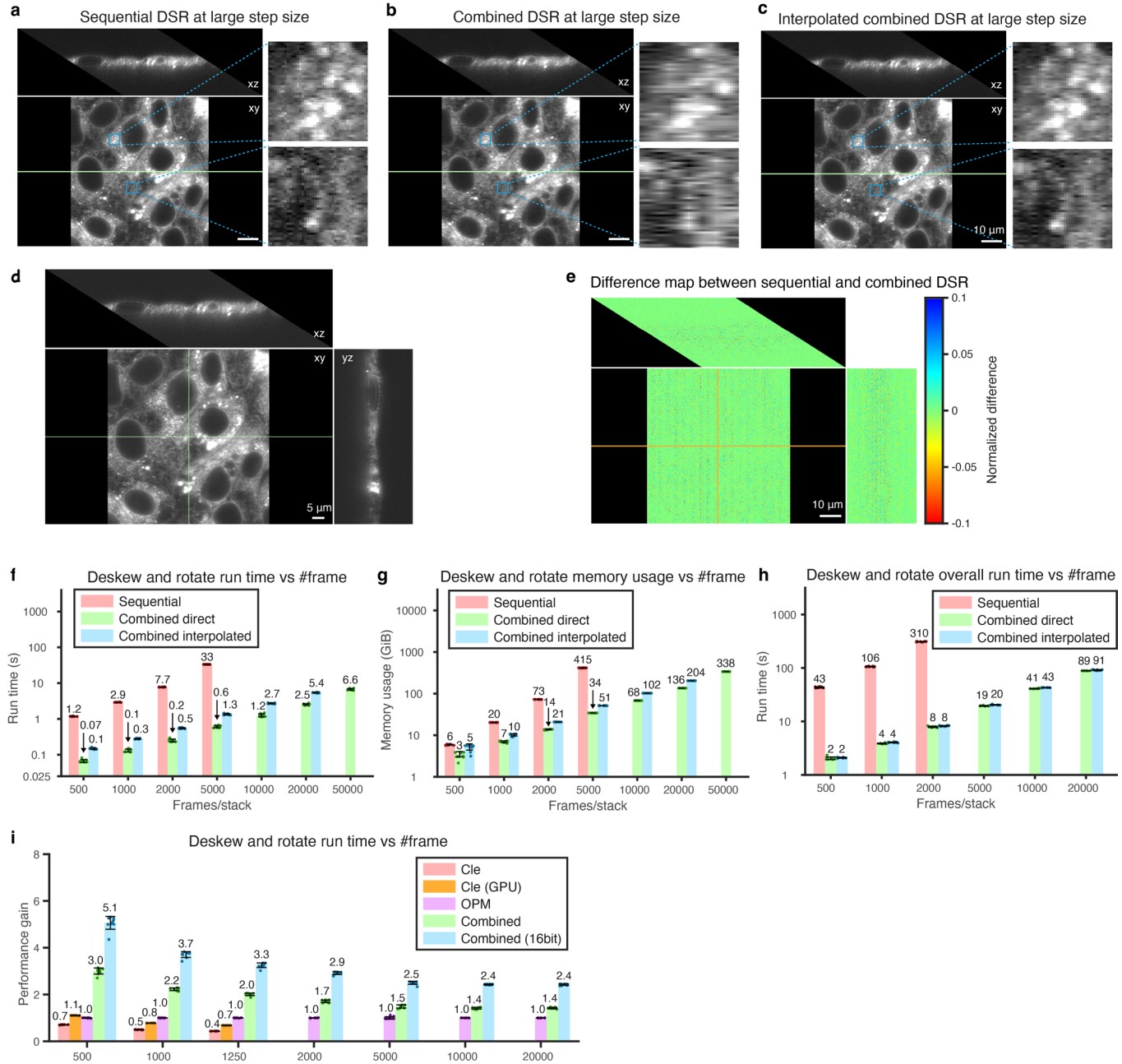

**Extended Data Fig. 2 | Additional benchmarks for deskew and rotation.**
**a**, sequential deskew/rotation for an image stack with a step size 0.6 μm between planes. **b**–**c**, combined deskew/rotation and interpolation plus combined deskew/rotation for the same image as in panel **a**. **d**, orthogonal views of combined deskew/rotation for the image in Fig. 3b. **e**, Difference map between sequential and combined deskew/rotation for the images in Fig. 3b and panel **d**. **f**–**h**, benchmarks of sequential, combined, and interpolated combined deskew/rotation versus the number of frames, for run time (**f**), memory usage (**g**), and overall run time including reading and writing (**h**). The benchmark results show the absolute run times and memory usages for Fig. 3f–h. In groups with larger frame numbers, some benchmarks (especially for the sequential method) failed due to out-of-memory issues and are not shown in the bar charts. **i**, benchmarks of the combined deskew/rotation methods implemented in pyclesperanto using both CPU ('Cle') and GPU ('Cle (GPU)'), qi2lab-OPM ('OPM'), and our combined interpolated approach ('Combined'). Results are normalized to the mean run times of the qi2lab-OPM method. The method in pyclesperanto failed for images with 1,500 or more frames. All benchmarks were performed with 32-bit float output. We also included results from our method with uint16 output ('Combined (16bit)'). For panels **f**–**i**, all images have a uint16 frame size of 512 × 1,800 (xy). Each benchmark was run independently ten times on a 24-core CPU computing node (dual Intel Xeon Gold 6146 CPUs), except for 'Cle (GPU)' which was run on a GPU node with 80 GB A100 GPUs. Data are shown as mean ± s.d. in panels **f**–**i**.

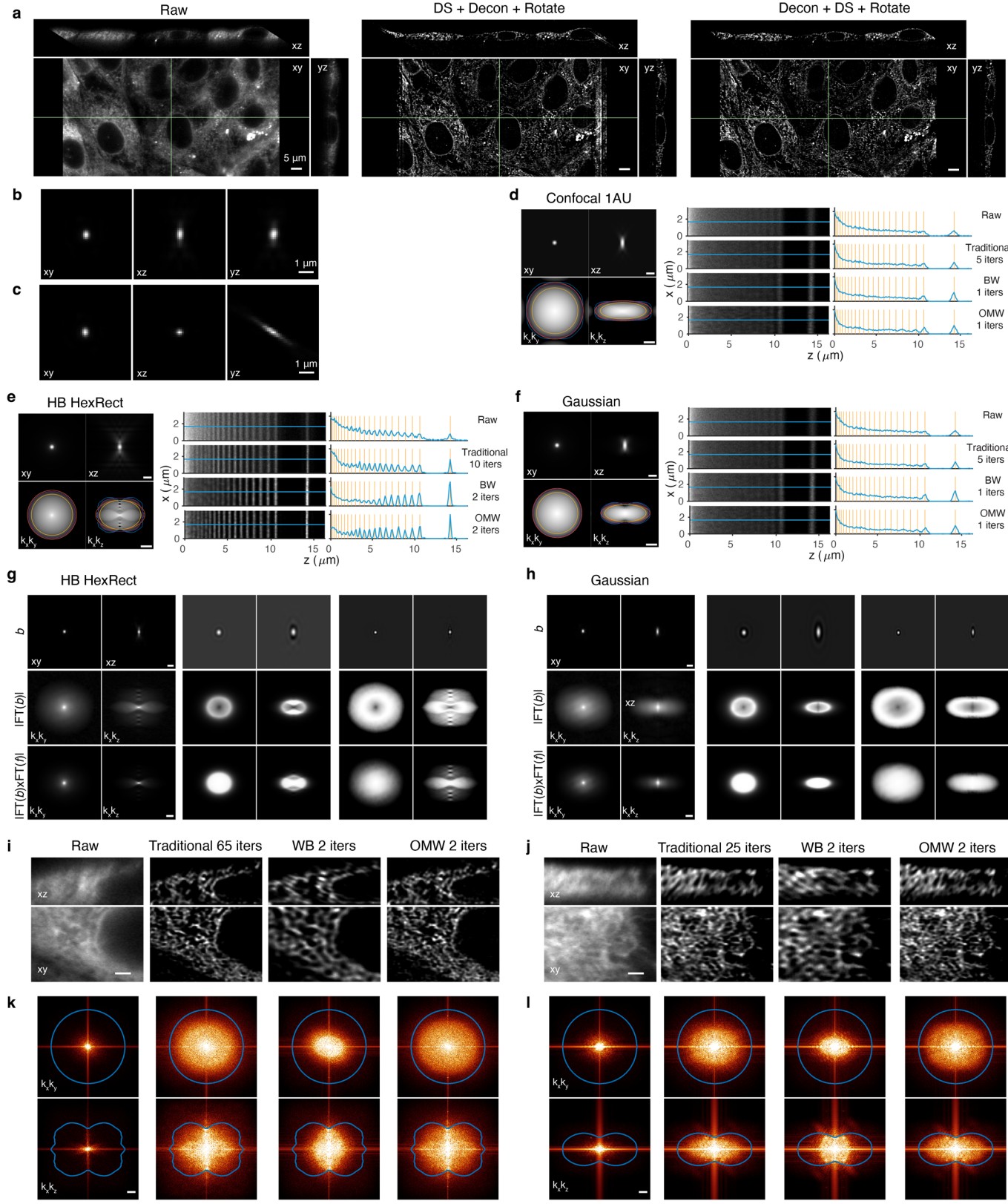

**Extended Data Fig. 3 | Comparison of deconvolution methods for different light sheets. a**, comparison of first deskew then deconvolution and rotation (center) versus first deconvolution then deskew and rotation (right), alongside the raw deskewed and rotated image (left). **b–c**, microscope PSF as seen in deskewed space (top) and skewed space (bottom). **d–f**, comparison of deconvolution of a simulated stripe pattern with different deconvolution methods for confocal with 1 Airy unit (AU) (**d**), harmonic- balanced (HB) HexRect light sheet (**e**), and Gaussian light sheet (**f**). **g–h**, illustration of backward projectors for different deconvolution methods for HB HexRect (**g**), and Gaussian light sheets (**h**). **i–j**, comparison of cell images deconvolved by different deconvolution methods for HB HexRect (**i**) and Gaussian (**j**) light sheets. **k–l**, Fourier spectra for the raw and deconvolved images in panels **i** and **j**.

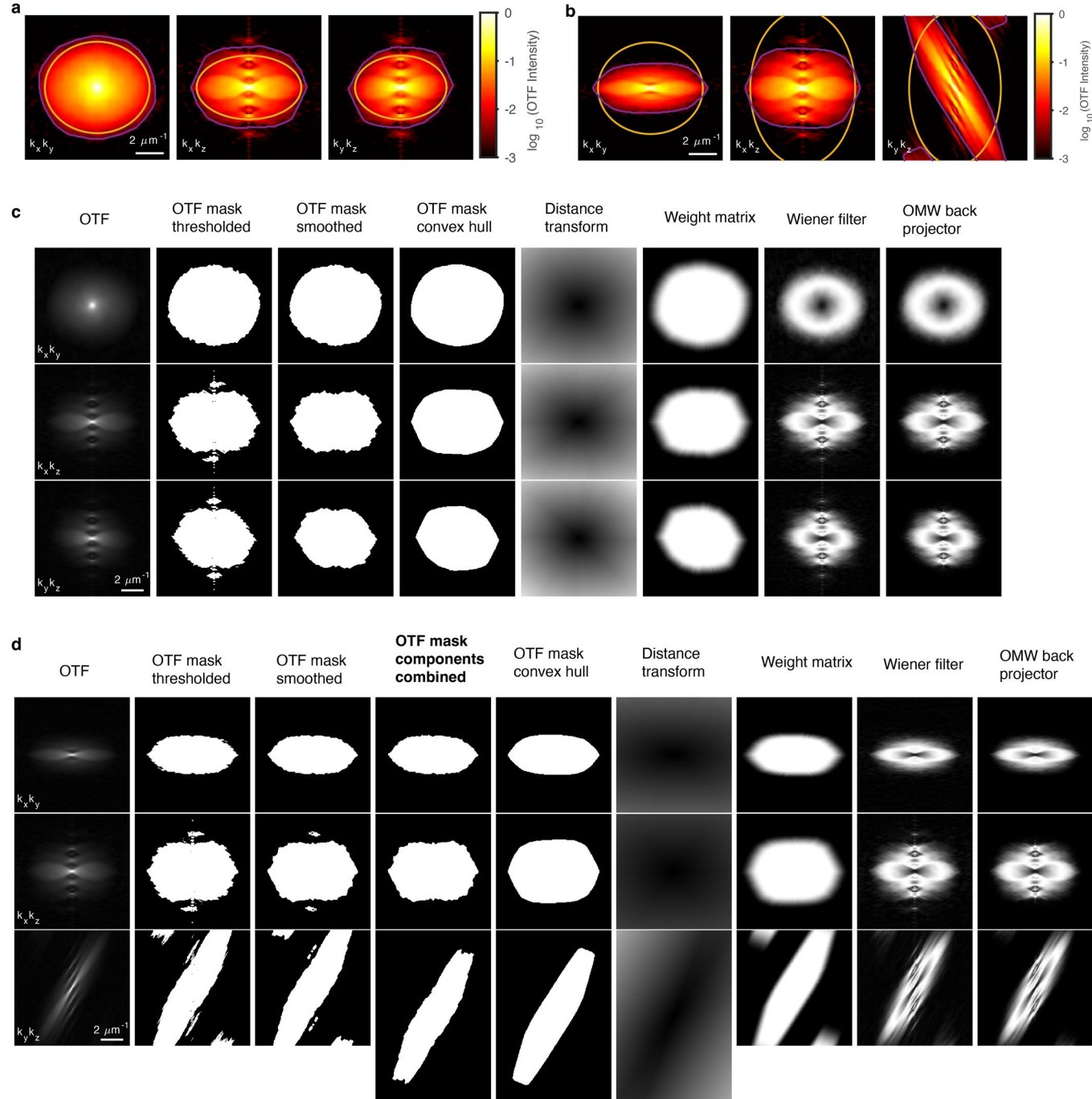

**Extended Data Fig. 4 | Generation of the OMW backward projector. a–b**, OTF support for WB (orange) and OMW (purple) methods for PSFs in deskewed (**a**) or skewed (**b**) spaces. **c**, OMW backward projector generation process using the PSF in the deskewed space. **d**, OMW backward projector generation process using the PSF in the skewed space.

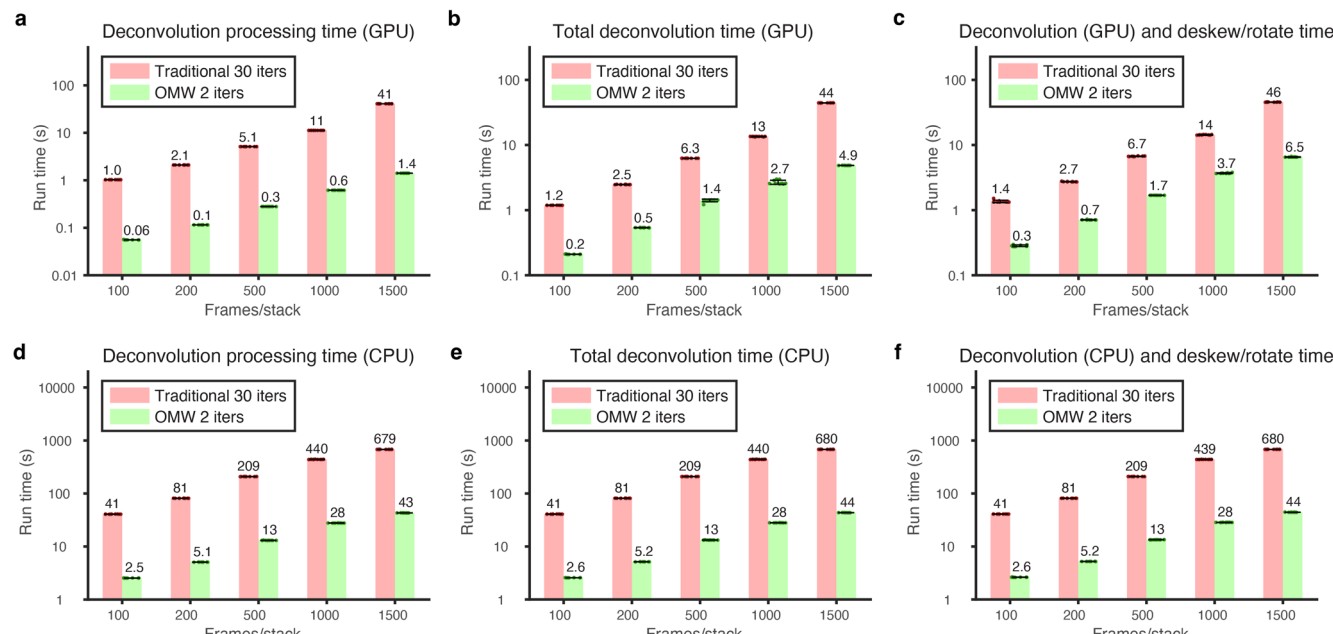

**Extended Data Fig. 5 | Additional benchmarks comparing conventional to OMW deconvolution. a** and **d**, deconvolution processing times only on GPU (**a**) and CPU (**d**). **b** and **e**, deconvolution plus read/write time on GPU (**b**) and CPU (**e**). **c** and **f**, deconvolution plus combined deskew/rotation time on GPU (**c**) and CPU (**f**). Each benchmark was run independently ten times on a 24-core CPU computing node (dual Intel Xeon Gold 6146 CPUs) for CPU benchmarks (**d**–**f**), and a GPU node with 80 GB A100 GPUs for GPU benchmarks (**a**–**c**). Data are shown as mean ± s.d. in all panels.

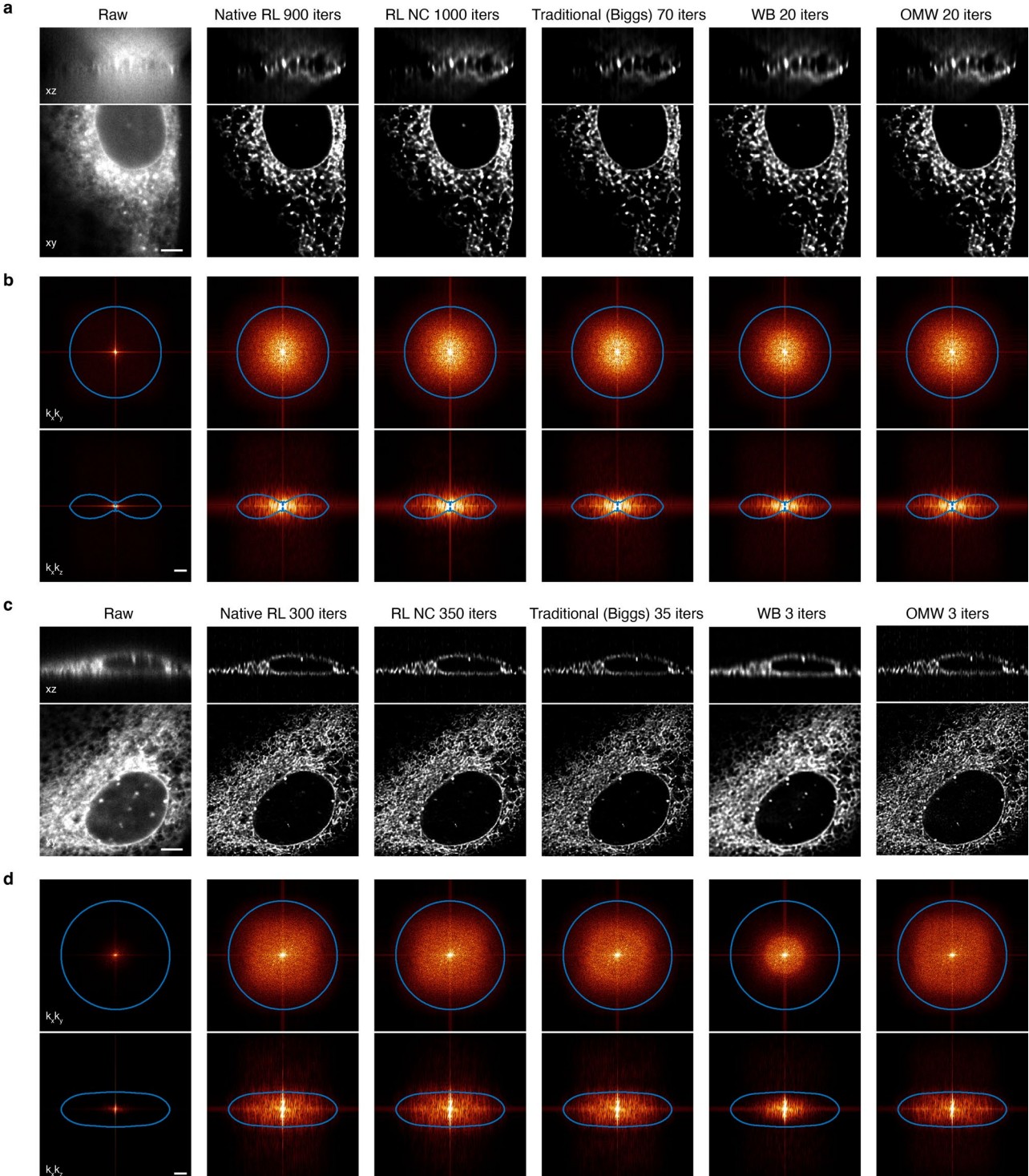

**Extended Data Fig. 6 | Deconvolution methods comparison for widefield and confocal microscopy images. a**, comparison of deconvolved orthoslices (scale bar: 5 μm) and **b**, Fourier spectra outputs for a widefield image (intensity γ = 0.5, scale bar: 1 μm⁻¹). **c**, comparison of deconvolved orthoslices (scale bar: 5 μm) and

**d**, Fourier spectra outputs for a confocal image (intensity γ = 0.5, scale bar: 1 μm⁻¹). 'RL NC' stands for the non-circulant RL method. The number of iterations is noted in the title for each method. For panels **b** and **d**, the blue lines indicate the theoretical OTF support. For panel **d**, the OTF support for 1 AU is shown.

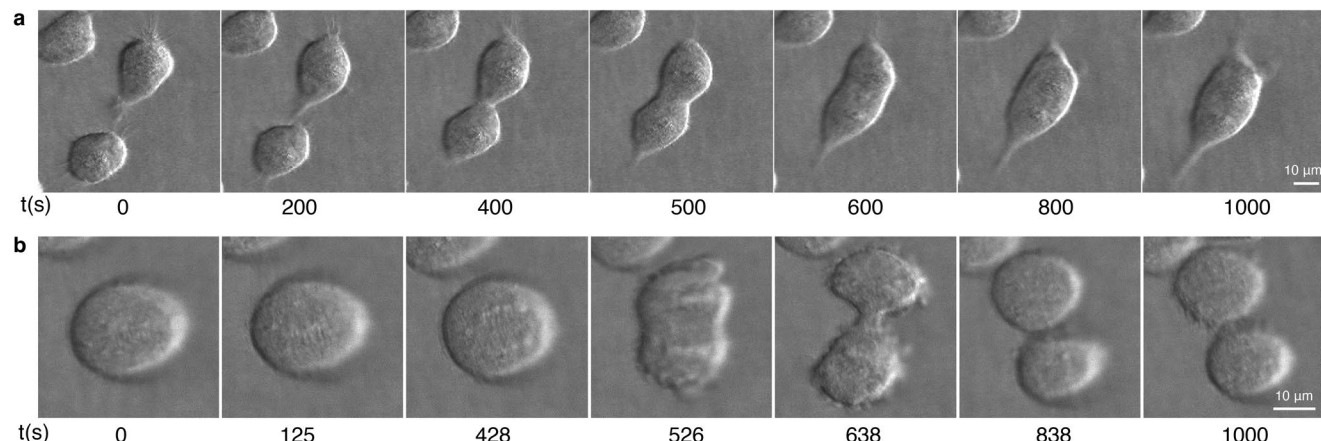

**Extended Data Fig. 7 | Time-lapse of cropped regions from the large field of view oblique illumination 'phase' imaging of HeLa cells in Fig. 5d and Supplementary Video 1. a**, the cropped regions for two cells merging over time. **b**, cropped regions for a dividing cell over time.

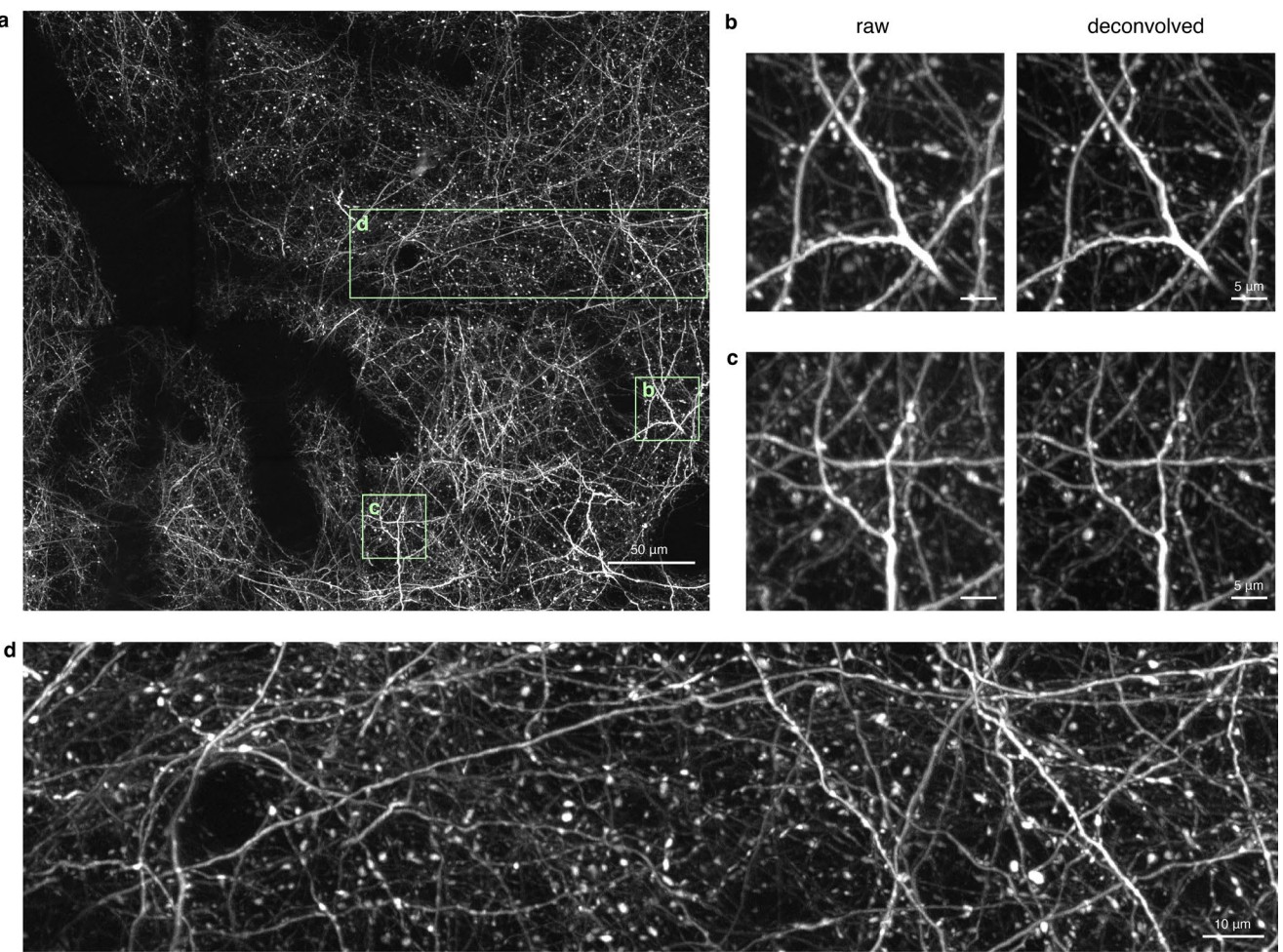

**Extended Data Fig. 8 | Application of the processing pipeline to two-photon live imaging of a mouse brain. a**, xy MIP of an adaptive optical two-photon image of dendrites and axons across a large stitched field of view in the cortex of a live mouse. **b**–**c**, comparison of raw and deconvolution images in smaller subregions. **d**, Example of an axon extending across 210 $\mu$m field of view.

# Reporting Summary

## Statistics

For all statistical analyses, confirm that the following items are present in the figure legend, table legend, main text, or Methods section.

| n/a | Confirmed | |
|-----|-----------|---|
| ☐ | ☒ | The exact sample size (*n*) for each experimental group/condition, given as a discrete number and unit of measurement |
| ☐ | ☒ | A statement on whether measurements were taken from distinct samples or whether the same sample was measured repeatedly |
| ☒ | ☐ | The statistical test(s) used AND whether they are one- or two-sided<br>*Only common tests should be described solely by name; describe more complex techniques in the Methods section.* |
| ☒ | ☐ | A description of all covariates tested |
| ☒ | ☐ | A description of any assumptions or corrections, such as tests of normality and adjustment for multiple comparisons |
| ☒ | ☐ | A full description of the statistical parameters including central tendency (e.g. means) or other basic estimates (e.g. regression coefficient) AND variation (e.g. standard deviation) or associated estimates of uncertainty (e.g. confidence intervals) |
| ☐ | ☒ | For null hypothesis testing, the test statistic (e.g. *F*, *t*, *r*) with confidence intervals, effect sizes, degrees of freedom and *P* value noted<br>*Give P values as exact values whenever suitable.* |
| ☒ | ☐ | For Bayesian analysis, information on the choice of priors and Markov chain Monte Carlo settings |
| ☒ | ☐ | For hierarchical and complex designs, identification of the appropriate level for tests and full reporting of outcomes |
| ☒ | ☐ | Estimates of effect sizes (e.g. Cohen's *d*, Pearson's *r*), indicating how they were calculated |

*Our web collection on statistics for biologists contains articles on many of the points above.*

## Software and code

Policy information about availability of computer code

| Data collection | LabView and Matlab |
|-----------------|---------------------|
| Data analysis | https://github.com/abcucberkeley/PetaKit5D<br>Matlab (R2023a, MathWorks), Python (3.8.8), Imaris (10.0, Oxford Instruments), Amira (2023.2, Thermo Fisher), Fiji (1.53t), and NVIDIA IndeX (NVIDIA), BaSiC (https://github.com/marrlab/BaSiC). |

For manuscripts utilizing custom algorithms or software that are central to the research but not yet described in published literature, software must be made available to editors and reviewers. We strongly encourage code deposition in a community repository (e.g. GitHub). See the Nature Portfolio guidelines for submitting code & software for further information.

## Data

Policy information about availability of data

All manuscripts must include a data availability statement. This statement should provide the following information, where applicable:
- Accession codes, unique identifiers, or web links for publicly available datasets
- A description of any restrictions on data availability
- For clinical datasets or third party data, please ensure that the statement adheres to our policy

Data will be made available upon reasonable request and the means of transfer provided (disks, globus, etc). Since the datasets are large (e.g., the VNC data 38 TiB and time-lapse live cell data 8.1 TiB) for any open repository, we uploaded representative subsets of the datasets to Dryad, and they can be accessed from: https://

## Human research participants

Policy information about studies involving human research participants and Sex and Gender in Research.

| | |
|---|---|
| Reporting on sex and gender | N/A |
| Population characteristics | N/A |
| Recruitment | N/A |
| Ethics oversight | N/A |

Note that full information on the approval of the study protocol must also be provided in the manuscript.

# Field-specific reporting

Please select the one below that is the best fit for your research. If you are not sure, read the appropriate sections before making your selection.

☒ Life sciences   ☐ Behavioural & social sciences   ☐ Ecological, evolutionary & environmental sciences

For a reference copy of the document with all sections, see nature.com/documents/nr-reporting-summary-flat.pdf

# Life sciences study design

All studies must disclose on these points even when the disclosure is negative.

| | |
|---|---|
| Sample size | No sample-size calculations were needed in this study. We used real or synthetic image data to run the computational benchmarks to demonstrate the performance of our software. |
| Data exclusions | No data were excluded in the manuscript. |
| Replication | No replicate imaging experiments were performed. For computational benchmarks, all replicates (typically 3-10) were performed independently and successful. The replication numbers are described in the manuscript. |
| Randomization | There were no experimental groups, so randomization was not applicable. |
| Blinding | There were no experimental groups, so blinding was not applicable. |

# Reporting for specific materials, systems and methods

We require information from authors about some types of materials, experimental systems and methods used in many studies. Here, indicate whether each material, system or method listed is relevant to your study. If you are not sure if a list item applies to your research, read the appropriate section before selecting a response.

## Materials & experimental systems

| n/a | Involved in the study |
|---|---|
| ☐ | ☒ Antibodies |
| ☐ | ☒ Eukaryotic cell lines |
| ☒ | ☐ Palaeontology and archaeology |
| ☐ | ☒ Animals and other organisms |
| ☒ | ☐ Clinical data |
| ☒ | ☐ Dual use research of concern |

## Methods

| n/a | Involved in the study |
|---|---|
| ☒ | ☐ ChIP-seq |
| ☒ | ☐ Flow cytometry |
| ☒ | ☐ MRI-based neuroimaging |

# Antibodies

| | |
|---|---|
| Antibodies used | Chicken anti-GFP (1:1000, Abcam, ab13970)<br>Rabbit anti-dsRed (1:1000, Takara Bio, 632496)<br>Goat anti-chicken IgY Alexa Fluor 488 (1:500, Invitrogen, A11039)<br>Goat anti-rabbit IgG Alexa Fluor 568 (1:500, Invitrogen, A11011) |
| Validation | Chicken anti-GFP: specific for GFP, validated for immunofluorescence from manufacturer: https://www.abcam.com/en-us/products/primary-antibodies/gfp-antibody-ab13970<br>Rabbit anti-dsRed: specific for dsRed, extensively validated for immunofluorescence with the summary and publications: https://www.takarabio.com/learning-centers/gene-function/fluorescent-proteins/fluorescent-protein-antibody-citations/rfp-antibody-citations |

# Eukaryotic cell lines

Policy information about cell lines and Sex and Gender in Research

| | |
|---|---|
| Cell line source(s) | LLC-PK1 cells were gifts from M. Davidson at Florida State University, originally obtained from ATCC.<br>HeLa (CCL-2) cells were purchased from ATCC. |
| Authentication | No further authentication was performed for this study. |
| Mycoplasma contamination | The cell lines were tested for mycoplasma contamination and the results were negative. |
| Commonly misidentified lines<br>(See ICLAC register) | None of the cell lines used here belongs to commonly misidentified lines. |

# Animals and other research organisms

Policy information about studies involving animals; ARRIVE guidelines recommended for reporting animal research, and Sex and Gender in Research

| | |
|---|---|
| Laboratory animals | Transgenic Thy1-YFP-H mice (The Jackson Laboratory) of 8 weeks or older; adult (female, 4-6 days after eclosure) fruit flies (Drosophila melanogaster). |
| Wild animals | No wild animals were used in this study. |
| Reporting on sex | Male or female mice; female fruit flies. |
| Field-collected samples | No field-collected samples were used in this study. |
| Ethics oversight | All mice experiments were conducted at Janelia Research Campus, Howard Hughes Medical Institute (HHMI) in accordance with the US National Institutes of Health Guide for the Care and Use of Laboratory Animals. Procedures and protocols were approved by the Institutional Animal Care and Use Committee of the Janelia Research Campus, HHMI. |

Note that full information on the approval of the study protocol must also be provided in the manuscript.

