## [Peer Review File · Nature Methods]

Image processing tools for petabyte-scale light sheet microscopy data

Corresponding Author: Professor Srigokul Upadhyayula

A version of this paper was originally rejected for publication by Nature Methods, however that decision was reconsidered after appeal by the authors.

Version 0:

Decision Letter:

10th Apr 2024

Dear Gokul,

Your Article entitled "Image processing tools for petabyte-scale light sheet microscopy data" has now been seen by three reviewers, whose comments are attached. While they find your work of potential interest, they have raised serious concerns which in our view are sufficiently important that they preclude publication of the work in Nature Methods, at least in its present form.

As you will see, the reviewers raise concerns about the benefits of the tool over existing software and whether LLSM5DTools is limited only to lattice light sheet data (vs LSFM data more broadly), in addition to more detailed technical questions about the performance and code.

Should further experimental data allow you to fully address these criticisms we would be most willing to look at a revised manuscript (unless, of course, something similar has by then been accepted at Nature Methods or appeared elsewhere). This includes submission or publication of a portion of this work somewhere else. We hope you understand that until we have read the revised paper in its entirety we cannot promise that it will be sent back for peer-review.

If you are interested in revising this manuscript for submission to Nature Methods in the future, please contact me to discuss your appeal before making any revisions. Otherwise, we hope that you find the reviewers' comments helpful when preparing your paper for submission elsewhere.

Sincerely,
Rita

Rita Strack, Ph.D.
Senior Editor
Nature Methods

Reviewers' Comments:

Reviewer #1:

Remarks to the Author:

A. The authors present software enabling more efficient storage, processing and visualization of large light sheet datasets. This work is important, as the authors point out, since even with the amalgam of existing tools, it is still slow and sometimes not possible to extract meaningful biological insights from these datasets. LLSM5Tools presents a nicely streamlined option with significant performance improvement.

B. The authors have developed novel reading/writing software for Tiff and Zarr formatted data which will be very useful. The assembly together of the LLSM efficient end-to-end preprocessing in a distributed computing environment is novel, and the RL deconvolution is clever.

C. Data is of high quality and presented clearly.

D. Statistics are appropriate.

E. The authors conclusions are sound. I think it would be good to mention the LLSM5Tools software doesn't enable registration/fusion of multiple volumes from different angular views (unless this is somehow doable with the stitching tool). It is also not clear how much the GUI has visualization built-in to it or how difficult it is to convert the Zarr store to the formats used in the InteX visualization. Being clear about these items will be helpful to the part of the light sheet community looking for those capabilities.

F. It could be good to provide more details on the comparative stitching so that the trials could be replicated. What parameters/settings were used in the cases of the different software? See also a few comments for consideration at the end of this alphabetical list.

G. The only reference that came to mind that the authors may want to consider was Terasstitcher. I'm not familiar with the tool, but several colleagues have said they find it useful.

H. Overall the writing is clear, though the introduction could be more concise and makes some redundant statements found also in the results regarding the challenges of light sheet datasets. The authors may consider, if space allows, giving a little more background on the computing concepts... for example many folks will understand why Matlab was chosen for this task... probably because of its great distributed computing toolbox, but most won't. Also, many readers won't be too familiar with the different .tiff specifications and that they have different limitations referred to in the manuscript. A small background statement in these areas and others could be helpful to make the manuscript more accessible.

Additional minor comments:

line 54 "explode to hundreds of gigabytes to petabytes." may be better as "explode to an upper limit in the petabyte range." in order to avoid the double "to".

line 58 consider removing "even" since it already appears earlier in the sentence.

line 61 change "hours rates too fast for conventional..." to "hours, rates too fast for conventional..."

line 72 sentence starts with dangling modifier "This often requires..." consider something like "This situation often necessitates..."

line 133 claims the authors have used OpenMP to read/write tiffs in multi-threaded fashion, but line 142 says tiffs are single-containers and so restricted to single-threaded operations. These two statements seem incompatible. Perhaps a little more explanation here (as in the discussion in the supplementary) would keep readers from getting confused.

Figure 1: Include enumeration of "OMW" in the legend, as most folks won't be familiar with the term "OTF Masked Wiener".

line 145 it's not clear what "border regions" is referring to, but I guess it's in case the array isn't an integer multiple of a chunk size, then there will be a "border" chunk? My understanding from the zarr specifications is this border will still be a full chunk size in storage, but the unused values should be set to something that makes sense.

line 521 I think "compiles" should be "complies"

Reviewer #2:

Remarks to the Author:

Managing the initial steps of treatment of large-scale image data is one of the bottleneck of 3D histological analysis, even before their actual processing (segmentation, reconstructions, analysis) has even started. Providing a framework to handle these tasks in distributed architectures in a highly optimized package is of great importance. While many of the developments proposed here, such as deskew, only apply to specific types of light sheet modalities, the fast I/O and stitching alone should be immensely useful for any imaging modalities producing large amounts of data. I am particularly interested in the fast Zarr implementation, and strategies to work with unconventional image shapes, without resorting to 0 padding.

Strengths:

- This manuscript describes a valuable set of computational tools for streamlining initial processing of large image datasets, particularly from lattice light sheet microscopy.
- The framework is designed for distributed computing and offers functionalities like conversion, stitching, deskewing, and deconvolution.
- The writing is clear and accessible to a broad scientific audience.

Major Revision:

The benchmark data used to showcase the tool's capabilities could be improved. While the focus is computational, the current data doesn't effectively demonstrate the potential for analyzing biological samples at this scale.

Consider including examples that illustrate the unique possibilities of large-scale biological imaging and highlight the limitations of simpler imaging modalities in addressing specific research questions (for instance, could 2D be sufficient to seek mitosis patterns in cultured cells?).

The point of this work is to show that it enables amazing novel applications relying on very large-scale data, but the biology, or quality of the data presented may not do justice to the capacities of such systems. In its current presentation, it can seem sometimes that this catering more to the needs arising from better technical capabilities, but not from the biological questions. I would strongly suggest 1) providing a better explanation and context for the data shown 2) seeking out other datasets that would illustrate the new frontiers opened by the capacity to process data of that size.

Suggestions for Improvement:

1. Conflict of Interest: Disclose any potential conflict of interest arising from an Nvidia scientist co-authoring the manuscript and the use of the Nvidia platform (i.e., Idex).
2. Tiff Readers: Briefly discuss the rationale behind using fast Tiff readers, acknowledging alternative approaches like memory mapping for specific use cases.
3. Live Stitching Clarification: Explain how stitching functions in "live" mode, considering the limitations of global optimization in this scenario.
4. Deconvolution Jargon: Simplify the deconvolution explanation in the main text, reserving technical details for the methods section.
5. Tool Applicability: Re-evaluate the title and tool descriptions to accurately reflect the broader applicability beyond lattice light sheet systems. For instance, include examples utilizing other light sheet microscopy techniques (e.g., commercial Gaussian systems).

Overall, I am excited for this line of work, and am looking forward to test it in house. The suggested revisions aim to strengthen the manuscript by showcasing the biological impact and ensuring broader applicability is clearly communicated.

Nicolas Renier

Reviewer #3:

Remarks to the Author:

Ruan et al. present a computational tool suite called LLSM5DTools for processing lattice light sheet microscopy (LLSM) data. These include a distributed computing framework, file I/O, deskew and rotation, deconvolution, and registration plus stitching. The manuscript's central claim is that these tools are more efficient than existing computational tools and may enable new discoveries for very large datasets.

The file I/O, deskew plus rotation, and deconvolution sections do not compare to state-of-the-art tools. Perhaps more critically, it is unclear how processing tools tied to the metadata and geometry of one style of microscope apply to the broader community.

The registration and stitching speed gains are exciting. If paired with a more general user interface for non-LLSM data or made easily compatible as a stitching background for existing tools, they could impact the overall community.

For file I/O, the Zarr Python package explicitly does not provide parallel read and write capability, only parallel (de-)compression. A more appropriate comparison would be Zarr paired with Dask, particularly the lightweight methods of such Dask Array's `map_blocks()`, to utilize a Zarr store as a sink for parallel writing. For the tiff file package, similar parallelism is achieved using Dask and yielded sub-blocks of a tiled OME-TIFF. How do the MPI-based packages compare to these properly parallelized approaches?

To avoid a Dask dependence, how does the performance compare to other microscopy-specific open-source parallelized Zarr writers, such as the acquire project: <https://github.com/acquire-project/acquire-driver-zarr> or [xarray-tensorstore](https://github.com/google/xarray-tensorstore): <https://github.com/google/xarray-tensorstore>

Combining deskew plus rotation is not new. Working with oblique plane data, the Dunsby group has used various orthogonal interpolation and combined matrix schemes to reduce the number of steps since at least 2016 (See V. Maioli's 2017 Ph.D. thesis for further discussion). Multiple implementations are available online, for example, at https://github.com/ciEsperanto/pyclesperanto_prototype or <https://royerlab.github.io/dexp/build/html/index.html>. The two packages above are meant for integrated large-scale processing and are not discussed in the manuscript.

The deconvolution tools are a further step in this group's effort to deal with deconvolving LLSM data (their previous publication is ref 19 in the manuscript). It is unclear how these tools compare to advanced RL implementations, such as RL with non-circulant edge handling. That said, the OTF masked Wiener technique appears to have value for LLSM, and further advancements in deconvolution will undoubtedly speed up the process of dealing with big data. Demonstrating that the proposed approach is helpful beyond LLSM data would make clear the approach should be adopted.

In my opinion, the registration and stitching tools are the most exciting part of the manuscript. Here, the authors do compare to many state-of-the-art tools, but they still miss a few, such as BigStream (<https://github.com/JaneliaSciComp/bigstream>). A more detailed discussion of the benchmarking approaches would help clarify ZarrStitcher's advantages. For example, BigStitcher-Spark can handle raw skewed data using transformations in the XML. Does this improve performance? How does chunk size affect performance for the respective packages?

Generally, I think the stitching and fusion performance claims in this part of the manuscript are exciting and could enable new science. I could not verify this performance using our own data using LLSM5DTools. I assume that is due to incorrect settings, as I do not usually use MATLAB. Some guidance on optimizing chunk sizes, overlaps, and other parameters will help guide software users and aid adoption. I also note I could not get the LLSM5DTools executable to run.

Visualization and real-time feedback tools, such as NVIDIA Index, are also discussed. The manuscript and GitHub contain insufficient information for a developer, let alone an end user, to utilize Index for real-time visualization.

I had to dig into the LLSM5DTools GitHub for all comparisons to find the settings and datasets used for all the benchmarks. These should be published with the manuscript or placed on Zenodo (or equivalent).

Overall, this paper has some exciting components, and I agree with the authors that efficient data processing is a major hurdle for extracting knowledge from multi-terabyte and petabyte-level imaging. However, the lack of comparisons, hard coding of the parameters and the GUI to lattice light sheet microscopy, and the overall lack of detail will limit LLSM5DTools' impact on the broader community.

** For Nature Portfolio general information and news for authors, see <http://npg.nature.com/authors>.

Version 1:

Decision Letter:

2nd May 2024

Dear Gokul,

Sorry again for our delay due to some team members being away from the office last week.

Thank you for your letter asking us to reconsider our decision on your Article, "Image processing tools for petabyte-scale light sheet microscopy data". After careful consideration we have decided that we are willing to consider a revised version of your manuscript that is updated as you've described.

My colleagues and I think it would be appropriate to benchmark as we discussed in our chat, specifically against tools that, as you say, function "within an order of magnitude" of LLSM5DTools in terms of dataset size or speeds.

We do not require new biological demonstrations, but please emphasize how the tool benefits the quality/throughput/scale of biological dataset analysis.

Please also emphasize that LLSM5DTools is a general suite for light sheet and not hard-coded for LLSFM.

Please also keep me in the loop about the Code Ocean version upon resubmission for the referees.

* include a point-by-point response to our referees and to any editorial suggestions

* please underline/highlight any additions to the text or areas with other significant changes to facilitate review of the revised manuscript

* address the points listed described below to conform to our open science requirements

* ensure it complies with our general format requirements as set out in our guide to authors at www.nature.com/naturemethods

* resubmit all the necessary files electronically by using the link below to access your home page

Link Redacted

We hope to receive your revised paper within three months. If you cannot send it within this time, please let us know. In this event, we will still be happy to reconsider your paper at a later date so long as nothing similar has been accepted for publication at Nature Methods or published elsewhere.

OPEN SCIENCE REQUIREMENTS

REPORTING SUMMARY AND EDITORIAL POLICY CHECKLISTS

When revising your manuscript, please submit reporting summary and editorial policy checklists.

DATA AVAILABILITY

CODE AVAILABILITY

Please include a "Code Availability" subsection in the Online Methods which details how your custom code is made available. Only in rare cases (where code is not central to the main conclusions of the paper) is the statement "available upon request" allowed (and reasons should be specified).

MATERIALS AVAILABILITY

ORCID

Nature Methods is committed to improving transparency in authorship. As part of our efforts in this direction, we are now requesting that all authors identified as 'corresponding author' on published papers create and link their Open Researcher and Contributor Identifier (ORCID) with their account on the Manuscript Tracking System (MTS), prior to acceptance. This applies to primary research papers only. ORCID helps the scientific community achieve unambiguous attribution of all scholarly contributions. You can create and link your ORCID from the home page of the MTS by clicking on 'Modify my Springer Nature account'. For more information please visit <http://www.springernature.com/orcid> or visit www.springernature.com/orcid.

Sincerely,
Rita

Rita Strack, Ph.D.
Senior Editor
Nature Methods

Author Rebuttal letter:

Dear Dr. Strack,

Thanks for the continued consideration of our manuscript "Image processing tools for petabyte-scale light sheet microscopy data". We thank the reviewers for their comments, which we address as follows:

The key changes are summarized below:

- Renamed the software from "LLS5D Tools" to "PetaKit5D" to highlight the general capabilities to process petabyte-scale 5D image data.
- Expanded the scope of the introduction and related results sections to provide a broader overview.
- Emphasized the processing of both non-lattice light sheet images and images from other imaging modalities.
- Included additional benchmarks for Tiff and Zarr readers and writers, deskew and rotation, deconvolution, and stitching.
- Improved documentation for Nvidia IndeX software access and usage with example code and data.
- Uploaded benchmark code on Zenodo and provided Code Ocean container for software demos and tests.
- Added a new Parallel Fiji Visualizer plugin for fast Tiff and Zarr file opening in ImageJ/Fiji.
- Improved the software documentation, stability and accessibility. Added python wrappers for major functions.

Below are our responses to the reviewer comments.

Reviewer #1:

'The authors present software enabling more efficient storage, processing and visualization of large light sheet datasets. This work is important, as the authors point out, since even with the amalgam of existing tools, it is still slow and sometimes not possible to extract meaningful biological insights from these datasets. LLSM5Tools presents a nicely streamlined option with significant performance improvement.'

'The authors have developed novel reading/writing software for Tiff and Zarr formatted data which will be very useful. The assembly together of the LLSM efficient end-to-end preprocessing in a distributed computing environment is novel, and the RL deconvolution is clever.'

'Data is of high quality and presented clearly.'

'Statistics are appropriate.'

'The authors conclusions are sound.'

'I think it would be good to mention the LLSM5Tools software doesn't enable registration/fusion of multiple volumes from different angular views (unless this is somehow doable with the stitching tool).'

- We revised the Discussion section to note the limitations of our registration and stitching tools.

'It is also not clear how much the GUI has visualization built-in to it or how difficult it is to convert the Zarr store to the formats used in the InteX visualization. Being clear about these items will be helpful to the part of the light sheet community looking for those capabilities.'

- Our data processing GUI (PetaKit5D-GUI) does not have built-in visualization capabilities.

- There are two visualization tools discussed in the revised manuscript:

o For visualizing data using Fiji/ImageJ, we added a plugin (Parallel Fiji Visualizer) that leverages our fast image reader and writer libraries and provides a user-friendly drag-and-drop feature to quickly open large compressed Tiff and Zarr images (Supplementary Note 3, Fig. S12).

o We added additional details on how to access, install, and convert data formats for NVIDIA IndeX. Additionally, we added a Zenodo link (<https://doi.org/10.5281/zenodo.12539579>) to scripts and instructions to scale the visualization using multiple GPUs in a single workstation, or across several nodes on a supercomputer (Supplementary Note 5, pages 4-6 in SI).

'It could be good to provide more details on the comparative stitching so that the trials could be replicated. What parameters/settings were used in the cases of the different software? See also a few comments for consideration at the end of this alphabetical list.'

- We added the parameters used for different stitching software in Table S2 and the caption.

- Additionally, we included links to the benchmark scripts on Zenodo to re-run these comparison benchmarks (<https://doi.org/10.5281/zenodo.12676473>).

'The only reference that came to mind that the authors may want to consider was TeraStitcher. I'm not familiar with the tool, but several colleagues have said they find it useful.'

- We added a reference to TeraStitcher (ref. 18). At present, BigStitcher (Spark version) is considered among the state-of-the-art stitching software tools for multi-terabyte datasets.

- According to Hörl et al. 2019, BigStitcher outperforms TeraStitcher (please see Supplementary Table 2 in their paper). Based on this, we found it reasonable to exclude TeraStitcher from our comparison benchmarks.

'Overall the writing is clear, though the introduction could be more concise and makes some redundant statements found also in the results regarding the challenges of light sheet datasets. The authors may consider, if space allows, giving a little more background on the computing concepts... for example many folks will understand why Matlab was chosen for this task... probably because of its great distributed computing toolbox, but most won't. Also, many readers won't be too familiar with the different .tiff specifications and that they have different limitations referred to in the manuscript. A small background statement in these areas and others could be helpful to make the manuscript more accessible.'

- Thank you for the suggestions. We have made the following changes to improve the clarity and conciseness of the introduction:

o Simplified the description of the software's main features, removing redundant background and technical details.

o Added a note on the software availability in MATLAB, Python, and GUI formats for a broader user community.

- We also added a brief description of different Tiff formats commonly used for microscopy data in the results section (First paragraph in Section 2.2).

'Additional minor comments:'

'line 54 "explode to hundreds of gigabytes to petabytes." may be better as "explode to an upper limit in the petabyte range." in order to avoid the double "to".'

- We have updated the text as suggested.

'line 58 consider removing "even" since it already appears earlier in the sentence.'

- We have updated the text as suggested.

'line 61 change "hours rates too fast for conventional..." to "hours, rates too fast for conventional..."'

- We have updated the text as suggested.

'line 72 sentence starts with dangling modifier "This often requires..." consider something like "This situation often necessitates..."'

- We have updated the text as suggested.

'line 133 claims the authors have used OpenMP to read/write tiffs in multi-threaded fashion, but line 142 says tiffs are single-containers and so restricted to single-threaded operations. These two statements seem incompatible. Perhaps a little more explanation here (as in the discussion in the supplementary) would keep readers from getting confused.'

- We revised the text to clarify the Tiff single-container limitation by noting 'only the compression process is parallelized during writing'.

'Figure 1: Include enumeration of "OMW" in the legend, as most folks won't be familiar with the term "OTF Masked Wiener".'

- We have revised the legend text as suggested.

'line 145 it's not clear what "border regions" is referring to, but I guess it's in case the array isn't an integer multiple of a chunk size, then there will be a "border" chunk? My understanding from the zarr specifications is this border will still be a full chunk size in storage, but the unused values should be set to something that makes sense.'

- We have revised this sentence for clarity.

'line 521 I think "compiles" should be "complies"'

- We have corrected this typo.

Reviewer #2:

'Remarks to the Author:

Managing the initial steps of treatment of large-scale image data is one of the bottleneck of 3D histological analysis, even before their actual processing (segmentation, reconstructions, analysis) has even started. Providing a framework to handle these tasks in distributed architectures in a highly optimized package is of great importance. While many of the developments proposed here, such as deskew, only apply to specific types of light sheet modalities, the fast I/O and stitching alone should be immensely useful for any imaging modalities producing large amounts of data. I am particularly interested in the fast Zarr implementation, and strategies to work with unconventional image shapes, without resorting to 0 padding.

Strengths:

- This manuscript describes a valuable set of computational tools for streamlining initial processing of large image datasets, particularly from lattice light sheet microscopy.
- The framework is designed for distributed computing and offers functionalities like conversion, stitching, deskewing, and deconvolution.
- The writing is clear and accessible to a broad scientific audience.'

'Major Revision:'

'The benchmark data used to showcase the tool's capabilities could be improved. While the focus is computational, the current data doesn't effectively demonstrate the potential for analyzing biological samples at this scale.

Consider including examples that illustrate the unique possibilities of large-scale biological imaging and highlight the limitations of simpler imaging modalities in addressing specific research questions (for instance, could 2D be sufficient to seek mitosis patterns in cultured cells?).'

- Analysis of biological specimens for new biological insights at the terabyte or larger scales is outside the scope of this specific work. We and other have demonstrated biological discoveries that were made using large volumetric datasets generated from high bandwidth microscopes.

- The work we present here is specific to data pre-processing and is independent of specific biological questions that requires subsequent and specialized analysis.

'The point of this work is to show that it enables amazing novel applications relying on very large-scale data, but the biology, or quality of the data presented may not do justice to the capacities of such systems. In its current presentation, it can seem sometimes that this catering more to the needs arising from better technical capabilities, but not from the biological questions.'

- Focusing in detail on a single biological question would not serve the broad interests of the readership or attract the attention of biologists researching other systems we wish to highlight. Our collaborators continue to analyze more extensive data we collected with them and processed using these tools, and their results should be published in due course.

- Throughout the manuscript, we include several examples that leverage the presented tools in the context of various biological experiments:

- o Large field-of-view (FOV) imaging of cells on coverslips using 2D oblique illumination (Fig. 5c-e and Fig. S9a-b)
- o Large FOV imaging using 3D LLSM (Fig. 5f-h) allowed us to see something unexpected – a 3-way cell division of LLC-PK1 cells (Fig. 5h, Movie S2 and S3) that are near impossible to catch with small field of view, slow, or 2D imaging.
- o We imaged and processed the entire fly ventral nerve cord (VNC) at 8x expansion (Fig. 6e-i and Movie S5).
- o In addition, where appropriate, we demonstrated our tools are general to a range of imaging modalities (various light sheets, widefield, confocal, phase, and two photon).

'I would strongly suggest 1) providing a better explanation and context for the data shown 2) seeking out other datasets that would illustrate the new frontiers opened by the capacity to process data of that size.'

- We have revised the manuscript to provide additional context and clarity for the data presented:

- Long-term time series cell data: We have expanded the discussion to emphasize how our tools facilitate practical exploration and investigation of cellular and subcellular dynamics in 4D (3D+time). This is now possible on tens or hundreds of terabytes of data collected over extended periods, enabling researchers to uncover new insights into long-term biological processes.

- Externally sourced ExA-SPIM dataset: We have added additional details on the ExA-SPIM dataset (~108 TiB). We demonstrate how our pipeline achieves a greater than 10x reduction in overall computing time while maintaining or improving the quality of the output compared to existing methods (Fig. S7). This highlights the practical impact of our tools on large-scale datasets.

'Suggestions for Improvement:'

'Conflict of Interest: Disclose any potential conflict of interest arising from an Nvidia scientist co-authoring the manuscript and the use of the Nvidia platform (i.e., Idex).'

- We have added a conflict of interest statement for the NVIDIA researchers and

disclosed the use of IndeX software created and maintained by NVIDIA.

'Tiff Readers: Briefly discuss the rationale behind using fast Tiff readers, acknowledging alternative approaches like memory mapping for specific use cases.'

- We added an explanation for fast Tiff readers and discussed the use case of memory mapping along with its drawbacks.

'Live Stitching Clarification: Explain how stitching functions in "live" mode, considering the limitations of global optimization in this scenario.'

- We revised the text to clarify that registration is not used in the live stitching workflow.

'Deconvolution Jargon: Simplify the deconvolution explanation in the main text, reserving technical details for the methods section.'

- We simplified the technical description of the deconvolution in the main text, retaining only the necessary details to help readers understand the novelty.

'Tool Applicability: Re-evaluate the title and tool descriptions to accurately reflect the broader applicability beyond lattice light sheet systems. For instance, include examples utilizing other light sheet microscopy techniques (e.g., commercial Gaussian systems).'

- While the initial motivation of the software package was to accelerate processing the data generated using [AO/Ex]-LLSM, we recognize and demonstrate the generality of this toolkit. Therefore, we renamed the software from "LLSM5DTools" to "PetaKit5D" to eliminate the impression that the software is only useful for LLSM data.
- Additionally, we updated the Introduction and Results sections to cover a broader range of light sheet types and imaging modalities. These revisions ensure that the title and descriptions accurately reflect the broader applicability of the tools, including examples utilizing other imaging modalities.

'Overall, I am excited for this line of work, and am looking forward to test it in house. The suggested revisions aim to strengthen the manuscript by showcasing the biological impact and ensuring broader applicability is clearly communicated.'

- We appreciate the feedback in spirit which it was intended. We believe our revised manuscript balances demonstrating the performance benchmarks of our toolkit and a range of biological applications for a broader audience.

Reviewer #3:

'Ruan et al. present a computational tool suite called LLSM5DTools for processing lattice light sheet microscopy (LLSM) data. These include a distributed computing framework, file I/O, deskew and rotation, deconvolution, and registration plus stitching. The manuscript's central claim is that these tools are more efficient than existing computational tools and may enable new discoveries for very large datasets.'

'The file I/O, deskew plus rotation, and deconvolution sections do not compare to state-of-the-art tools. Perhaps more critically, it is unclear how processing tools tied to the metadata and geometry of one style of microscope apply to the broader community.'

- We have revised the manuscript to address the concerns about the comparison to state-of-the-art tools and the applicability of our tools to a wider range of microscopy techniques.

- We expanded the benchmarks and comparisons to provide a more comprehensive assessment of our tools' performance relative to existing state-of-the-art alternatives. This includes direct comparisons in terms of speed, accuracy, and memory usage, where applicable.

- We revised the scope of the manuscript to showcase the versatility of our toolkit beyond lattice light sheet microscopy. Our toolkit is extremely configurable and consequently applicable to various types of microscopes – both commercial and bespoke.

- To illustrate the flexibility of PetaKit5D, we added a document that lists all configurable input parameters within major functions on GitHub:
o https://github.com/abcucberkeley/PetaKit5D/blob/main/major_functions_documentation.txt

'The registration and stitching speed gains are exciting. If paired with a more general user interface for non-LLSM data or made easily compatible as a stitching background for existing tools, they could impact the overall community.'

- In the revised version, we focus on the generality of using these stitching tools using non-LLSM data. We include stitching examples for the following non-LLSM imaging modalities:

- o Two-photon microscopy: (Fig. S9c-f and Movie S4)
- o Oblique illumination microscopy: (Fig. 5c-e and Movie S1)
- o ExA-SPIM: (a light-sheet microscopy variant from the Allen Institute, Fig. S7)

- We revised the text to clarify the necessary data and parameters that are needed for successful stitching across different imaging modalities. Our GitHub repository documentation and stitching demo scripts

(https://github.com/abcucberkeley/PetaKit5D/blob/main/demos/demo_zarr_stitching.m and https://github.com/abcucberkeley/PetaKit5D/blob/main/demos/demo_phase_and_2photo_n_stitching.m) have been updated to instruct on specific options and parameters that may be unique to or required for different imaging modalities.

'For file I/O, the Zarr Python package explicitly does not provide parallel read and write capability, only parallel (de-)compression. A more appropriate comparison would be Zarr paired with Dask, particularly the lightweight methods of such Dask Array's `map_blocks()`, to utilize a Zarr store as a sink for parallel writing. For the tiff file package, similar parallelism is achieved using Dask and yielded sub-blocks of a tiled OME-TIFF. How do the MPI-based packages compare to these properly parallelized approaches?'

- We excluded Tiff/Zarr paired with Dask from our benchmarks, and do not agree it is an apt comparison to our I/O library. However, we performed the requested benchmarks:

- o Dask excels at combining multiple processing steps (e.g., reading, decompression, processing steps 1 to n, compression, writing) for multiple files (or small regions in a large array). Using Dask solely for reading or writing of a single file is not the optimal use case. When Dask is used just for these tasks, it repeatedly calls the same libraries (such as `tiffio` or `zarr-python`) in parallel, which introduces unnecessary overhead.
- o Our Cpp-Zarr library can read/write, (de)compress, and generate an assembled volume/array of the requested size. When Zarr paired Dask is tasked to do

exactly the same, we see little to no difference in performance compared to the native Zarr reader/writer. That is, Cpp-Zarr is 5-8x and 5-7x faster than Zarr paired with Dask for reader and writer, respectively.

- o When Zarr paired Dask reads and generates a NumPy list of data chunks without assembling these chunks into a single 3D volume, Cpp-Zarr is still ~ 30-50% faster.
- o Since a Tiff file is a single container, reading it with Dask paired tiffle library requires reading slices in parallel. This introduces additional overhead compared to using tiffle to read the entire volume at once, and can be slower. Additionally, writing multiple frames to a single Tiff file in parallel is not possible, however Cpp-Tiff enables parallel compression and serial writing, something that is not feasible with Dask.

- We find it appropriate to not include these comparisons in the revised manuscript since the intended use of Zarr paired Dask and Cpp-Zarr are not aligned. We have, however, included the benchmarking scripts and output figures in Zenodo (<https://doi.org/10.5281/zenodo.12676473>) to satisfy the reviewer's curiosity.

'To avoid a Dask dependence, how does the performance compare to other microscopy-specific open-source parallelized Zarr writers, such as the acquire project: <https://github.com/acquire-project/acquire-driver-zarr> or xarray-tensorstore: <https://github.com/google/xarray-tensorstore>'

- The benchmark against TensorStore to read and write Zarr data was included in the supplement (Table S1). We noted the performance of the preferred orders (C vs F for Python and MATLAB, respectively).
- Based on our discussions with Nathan Clark, one of the software developers on the Acquire Project, there is no practical way currently to call the Zarr reader and writer to benchmark against our file I/O library.

'Combining deskew plus rotation is not new. Working with oblique plane data, the Dunsby group has used various orthogonal interpolation and combined matrix schemes to reduce the number of steps since at least 2016 (See V. Maioli's 2017 Ph.D. thesis for further discussion). Multiple implementations are available online, for example, at https://github.com/ciEsperanto/pyclesperanto_prototype or <https://royerlab.github.io/dexp/buil/html/index.html>. The two packages above are meant for integrated large-scale processing and are not discussed in the manuscript.'

- We have revised the manuscript to acknowledge and cite the relevant contributions from the Dunsby group and others who have explored combined deskew and rotation approaches (refs. 9, 10, and 34).
- While we conceptually agree that combining deskew plus rotation is not a new idea, our implementation explores a different approach:
 - o The previous methods try to directly interpolate the voxels in the deskew/rotated space from the raw skewed data using customized interpolation methods.
 - o While both approaches belong to orthogonal interpolation, we interpolate in the raw skewed space to reduce effective scan step size instead of the final deskew/rotated space.
 - o Our method leverages the standard affine transformation for such combination, and we determined the condition for direct or interpolated processing. There are two major advantages of our approach:
 - § standard affine transformations are more efficient than customized orthogonal interpolation.
 - § can be directly combined with other user-defined affine transformations, i.e., resampling, cropping, other rotations, while previous approaches must first compute the deskew/rotated results before other affine transformations (some implementations include certain types of resampling, but integrating arbitrary rotations may be challenging).
- We also added a figure panel comparing our implementation with pyclesperanto_prototype (from ciE) (Fig. S2i).
 - o On CPU and for small data sizes (512 x 1,800 x 500 32-bit voxels or smaller), pyclesperanto_prototype performed marginally better than our implementation.
 - o For larger data sizes (500 to ~1,000 frames), our implementation was ~2x faster.
 - o The GPU implementation of pyclesperanto_prototype (up to 1250 frames) outperforms our CPU implementation, as expected.
 - o However, neither the CPU nor GPU implementation of pyclesperanto_prototype were able to process 1500 or more frames on a computing node with 512 GB RAM, and equipped with 80 GB NVIDIA A100 GPU (for GPU benchmarks). In contrast, our implementation successfully processed 20,000 frames on the same node.
- We also added a referenced to Royer lab's dexp software and noted that our combined deskew and rotation implementation is more than 50x faster on CPU.
- To the best of our knowledge, neither pyclesperanto_prototype nor dexp currently supports petabyte-scale processing in a multi-node setting, a key advantage of our implementation.

'The deconvolution tools are a further step in this group's effort to deal with deconvolving LLSM data (their previous publication is ref 19 in the manuscript). It is unclear how these tools compare to advanced RL implementations, such as RL with non-circulant edge handling. That said, the OTF masked Wiener technique appears to have value for LLSM, and further advancements in deconvolution will undoubtedly speed up the process of dealing with big data. Demonstrating that the proposed approach is helpful beyond LLSM data would make clear the approach should be adopted.'

- In the initial submission, we compared the OTF masked Wiener approach for both LLSM and traditional Gaussian light sheet in the manuscript (Fig. S3 h, j, l).
 - o Based on the reviewers' feedback, we added comparisons with other imaging modalities, i.e., widefield and confocal images, to demonstrate that our approach is significantly faster and generalizable to other imaging modalities.
- The traditional (conventional) RL method mentioned in the manuscript is not the native RL method, but the Biggs accelerated version of the RL method (refs. 39 and 63 in the manuscript), which is significantly faster than the native RL method.
 - o We have noted this in the method section (Visualization and software, lines 933-936), and we clarified this in the main manuscript and the related figures.
- We added a note (Supplementary Note 4) to discuss different RL methods. In our opinion, RL with non-circulant edge handling is a way to reduce edge artifacts. This approach has similar deconvolution performance when the edge artifacts are not of concern (i.e., no signal close to the volume edge, or large trimmable overlaps). It is not an exclusive method, but a feature that can be incorporated into other RL methods.
- We compared our methods with non-circulant RL and native RL methods for widefield and confocal images. As expected, our method achieves similar reconstruction performance with two orders of magnitude faster than these two methods (Fig. S6).

'In my opinion, the registration and stitching tools are the most exciting part of the manuscript. Here, the authors do compare to many state-of-the-art tools, but they still miss a few, such as BigStream (<https://github.com/JaneliaSciComp/bigstream>). A more detailed discussion of the benchmarking approaches would help clarify ZarrStitcher's advantages. For example, BigStitcher-Spark can handle raw skewed data using transformations in the XML. Does this

improve performance? How does chunk size affect performance for the respective packages?

- BigStream is a 3D registration framework, and its demonstrated use cases are to register and fuse pairs of volumes. To the best of our knowledge, it does not directly support registration of thousands of tiles without a fair bit of work to enable it. Although, it is doable pair by pair, it will be very inefficient.
- In addition to benchmarking against the original 'BigStitcher-Spark', we added benchmarks for the recently updated (currently unpublished) 'New BigStitcher-Spark' that contains substantial updates compared to the original. We consulted with Stephan Preibisch on the knobs to tune the performance of the 'New BigStitcher-Spark' for an apt comparison.
 - o ZarrStitcher is ~2-3x faster than 'New BigStitcher-Spark'.
 - o We added the parameter for the comparison in Table S2 and the caption.
 - o We compared the fusion performance for ExA-SPIM dataset, where the raw data is in the Cartesian coordinates (i.e., no need for deskewing or rotation).
 - o A reasonable chunk size within the range of 1283-5123 voxel does not significantly affect the performance of our stitcher. However, it may have an impact on the performance of the original BigStitcher-Spark. To ensure fairness in our comparison, we kept the same chunk sizes in the comparison.
 - o In the updated version of 'New BigStitcher-Spark', a new parameter called blockScale has been introduced, akin to our batch size parameter. We have explored this parameter and reported the results with the optimal parameters in Table S2.

'Generally, I think the stitching and fusion performance claims in this part of the manuscript are exciting and could enable new science. I could not verify this performance using our own data using LLSM5DTools. I assume that is due to incorrect settings, as I do not usually use MATLAB. Some guidance on optimizing chunk sizes, overlaps, and other parameters will help guide software users and aid adoption. I also note I could not get the LLSM5DTools executable to run.'

- We revised the repository to add demos, documentation, and refined the codebase to make it clearer and easier to use one's own data for stitching. We also created Python wrappers for the major functions and provided examples for users within the Python community. We expect this repository to improve with user/community feedback.
- We updated the GUI, simplified the installation process, improved the documentation, and provided several examples on the GitHub Wiki page.
- As part of this revised resubmission, we tested and included a CodeOcean container with the demo data and scripts to test all major tools described in the paper.

'Visualization and real-time feedback tools, such as NVIDIA Index, are also discussed. The manuscript and GitHub contain insufficient information for a developer, let alone an end user, to utilize Index for real-time visualization.'

- We updated the Supplementary Note 5 to include additional details on installation, configuration, and example use cases of NVIDIA Index.
- We included an example dataset, scripts, and scenes in Zenodo (<https://doi.org/10.5281/zenodo.12539579>), along with a copy of the NVIDIA Index software in the review files. The software is only for reviewer evaluation. Due to NVIDIA's policies, we cannot directly open source NVIDIA Index software. Interested users can obtain the software by following the instructions in Supplementary Note 5.

'I had to dig into the LLSM5DTools GitHub for all comparisons to find the settings and datasets used for all the benchmarks. These should be published with the manuscript or placed on Zenodo (or equivalent).'

- We prepared the testing workflow in CodeOcean to simplify initial evaluations. We have put the benchmarking code and datasets (where feasible) in Zenodo (<https://doi.org/10.5281/zenodo.12676473>, <https://doi.org/10.5281/zenodo.10471978>, and <https://doi.org/10.5281/zenodo.11500862>).

'Overall, this paper has some exciting components, and I agree with the authors that efficient data processing is a major hurdle for extracting knowledge from multi-terabyte and petabyte-level imaging. However, the lack of comparisons, hard coding of the parameters and the GUI to lattice light sheet microscopy, and the overall lack of detail will limit LLSM5DTools' impact on the broader community.'

- We included relevant comparisons in the initial submission either against the state-of-the-art, or tools that are within an order of magnitude in current performance for file IO, deconvolution and stitching.
 - o Based on the reviewers' comments, we added additional comparisons.
- The parameters of PetaKit5D are fully configurable. A document describing the configurable parameters, and the default values (where the user does not explicitly define the parameters) can be perused here:
 - o https://github.com/abcucberkeley/PetaKit5D/blob/main/major_functions_documentation.txt
 - o The parameters are not hard-coded in the software (including the Python wrappers and the GUI), rather the default parameter values are set in the input parsers (<https://www.mathworks.com/help/matlab/ref/inputparser.html>). These default values can be overridden with user-defined parameters during the function call.
 - o Users are encouraged to update the parameter values based on their use case.
- We made a concerted effort to demonstrate broader applicability of our toolkit. We added additional documentation and demos for other imaging modalities: 2-photon, confocal, phase, and widefield.

We hope that these revisions have adequately addressed the reviewers' concerns. Thank you very much for your consideration.

Sincerely,
Eric, Gokul & Xionghao

Version 2:

Decision Letter:

Our ref: NMETH-A55386B

13th Aug 2024

Dear Gokul,

Thank you for submitting your revised manuscript "Image processing tools for petabyte-scale light sheet microscopy data" (NMETH-A55386B). It has now been seen by the original referees and their comments are below. The reviewers find that the paper has improved in revision, and therefore we'll be happy in principle to publish it in Nature Methods, pending minor revisions to satisfy the referees'

final requests (as we discussed and reviewer 3 approved) and to comply with our editorial and formatting guidelines.

For completeness, please submit a point-by-point rebuttal outlining the changes when you resubmit.

TRANSPARENT PEER REVIEW

Please note: we allow redactions to authors' rebuttal and reviewer comments in the interest of confidentiality. If you are concerned about the release of confidential data, please let us know specifically what information you would like to have removed. Please note that we cannot incorporate redactions for any other reasons. Reviewer names will be published in the peer review files if the reviewer signed the comments to authors, or if reviewers explicitly agree to release their name. For more information, please refer to our <https://www.nature.com/documents/nr-transparent-peer-review.pdf> target="new">FAQ page.

ORCID

Sincerely,
Rita

Rita Strack, Ph.D.
Senior Editor
Nature Methods

Reviewer #1 (Remarks to the Author):

The authors have adequately addressed my concerns and have improved the manuscript clarity and code/benchmark documentation. I find the revised manuscript suitable for publication.

One minor comment on line 52, "these techniques have been used to millimeter-scale or larger..." should be "these techniques have been used to image millimeter-scale or larger..."

Reviewer #2 (Remarks to the Author):

The revised manuscript addresses many of the suggestions raised. The main text reads much better for a broad audience.

The authors have chosen not to implement my main suggestion, which was to demonstrate better use cases for extremely large-scale 3D-4D datasets. While I am still of the opinion that the biology shown here is not showcasing the power of the technology, the point is well taken that the goal of this manuscript is to present a technical pre-processing pipeline that is independent of specific applications. It may be that upcoming publications will demonstrate more specifically what can be gained by imaging in the scale of the hundreds of terabytes.

Nevertheless, this is an important set of tools that will facilitate large-scale imaging work, and I am very supportive of its publication in this journal.

Reviewer #2 (Remarks on code availability):

The code requires specific server architecture that I don't have access to yet, but I am definitely planning to make use of at least some of it in the near future.

Reviewer #3 (Remarks to the Author):

I have reviewed Ruan et al.'s revised manuscript describing a software kit now called PetaKit5D. The response letter and revised manuscript contain significant work to address the reviewer's comments. The authors now provide a set of results for various imaging modalities. Additionally, they now provide a clearer set of demo data, benchmarks, and code for evaluation.

I have two major concerns after further testing of the software (see code review). Before I discuss those, I want to commend the authors for making a large amount of their microscope data available for testing and for their efforts to provide examples of how to run the code on their microscope data.

1. How to integrate PetaKit5D into existing microscope control pipelines is unclear.

For example, how should someone change the function call in the PyPetaKit5D wrapper to load a .zarr file instead of a .tiff? I looked at the MATLAB code where the image paths are parsed and found that a suffix check (https://github.com/abucuberkeley/PetaKit5D/blob/172e065082ccb45158da14294c5a31406a65a338/microscopeDataProcessing/stitch/stitch_parse_image_list_information.m#L43), so it should be possible to pass a Zarr array per image file. However, that was not clearly explained in the documentation for the main stitching function (https://github.com/abucuberkeley/PetaKit5D/blob/172e065082ccb45158da14294c5a31406a65a338/major_functions_documentation.txt#L182). Given the authors make several claims about speeding up existing workflows, it would be helpful to have more explicit instructions on how to use the various functions when the data format on disk does not match the authors' format.

2. Many labs do not have access to such impressive computing resources. In my testing on a relatively high-end single-seat workstation in our lab, I could not run some of the benchmarks on the VNC data provided by the authors. I did run all of the benchmarks for the live cell data. The folder for the VNC data is ~350 GB, well short of the petabyte-level processing that the manuscript focuses on. Additionally, I found that some existing community tools were faster and could be completed on this workstation. Please see the code review section for more details. I think that addressing the minimum computer hardware necessary to use these tools at the terabyte and petabyte is critical to substantiate the claims in the manuscript.

Reviewer #3 (Remarks on code availability):

I locally installed the code on one of our Linux workstations (AMD Threadripper 12-core/24-thread, 128-GB RAM, Nvidia 4090 RTX, Ubuntu 22.04 LTS, 100 TB RAID0 SSD array) using the Python bindings. I made a Python 3.11 mamba environment for testing and installed only PyPetaKit5D in the environment, following the instructions.

DSR

I first downloaded the VNC data supplied on Zenodo. I made a separate directory with just one tile (000x_004y_001z_0000t.tif) for both CAMA and CAMB. I also copied all of the relevant metadata files for the utilized tile to that directory.

I tried the DSR example notebook and could not get it to run as-is. The code ran out of system memory and killed the kernel. Addressing the computational needs relates to my above questions about how microscopy labs with decent hardware, but not a cluster, can use this code.

I then set the `largeFile` flag to True and re-ran the code. The code gave an error,
"Error using XR_deskewRotateZarr

The input zarr file
/mnt/data/petakit5d_review/20220131_Korra_ExM_VNC_2ndtry/oneData/Scan_iter_0000_CamA_ch0_CAM1_stack0000_488nm_0000000msec_0013088542msecAbs_000x_004y_001z_0000t.tif does not exist!"

I ensured the `zarrFile` flag was False, so I was unsure how to proceed.

In contrast, the Numba-accelerated orthogonal interpolation code from doi: 10.1073/pnas.2220033120 ran on each CAMA and CAMB with an average of ~24s to load the TIFFs and ~39s for interpolation + deskew + rotation (10 runs after warming up the Numba function with a test call). I did not test writing for this file.

I then downloaded the live-cell data since that is smaller. Here, the example code ran and took ~5s per deskew to write the TIFF + MIP. These are extremely impressive file-writing times!

The orthogonal deskew combined with tiffio.imwrite took ~14s (average of 10 runs). The actual deskew function was ~1.3s of that, so I then tested with Tensorstore writing to zarr v2 with the same compression settings outlined in the manuscript. Using Tensorstore, with each z plane being one chunk, it drops the combined time to ~10s, further highlighting the impressive performance of the multi-threaded "cpp" image writers.

Stitching

folder for the VNC data is ~350 GB, well short of the petabyte-level processing that the manuscript focuses on. Additionally, I found that some existing community tools were faster and could be completed on this workstation. Please see the code review section for more details. I think that addressing the minimum computer hardware necessary to use these tools at the terabyte and petabyte is critical to substantiate the claims in the manuscript.'

We understand the importance of addressing the hardware requirements for using our toolkit effectively. We documented the hardware configurations used for single workstation and multi-node clusters used for developing, testing, and benchmarking PetaKit5D. All testing was performed on workstations with 512 GB memory. To address the reviewer's comment, we have added recommended computing resources to the readme in the GitHub page, while emphasizing that these are guidelines, as exact minimum requirements will vary on the data and specific workflows.

The toolkit is mainly designed to enable the scale of projects that are impractical with typical institutional computing resources, aiming to process hundreds of terabytes to petabyte-scale datasets. While it is certainly possible to use these tools on workstations with fewer resources, these limitations must be considered when assembling a pipeline. We have updated the demo (demo_large_scale_processing.m) to recommend that users with limited computing resources (facing out-of-memory issues) first convert their images to Zarr format, and then use large-scale processing frameworks for further processing, i.e., setting the 'zarrFile' and the 'largeFile' parameters to true, and changing 'batchSize' accordingly.

For deskew and rotation operations where resampling is unnecessary, we implemented a new method that performs the deskew/rotation as a 2D transformation in the xz plane. Additionally, we optimized the skewed space stitching function to reduce computational overhead and implemented the 2D transformation for deskew/rotation using SIMD programming techniques. This new implementation significantly outperforms the reviewer referenced Cle (both CPU and GPU) and OPM methods in terms of speed (Extended Data Fig. 2i). We also added a feature that allows for direct uint16 input and output while performing calculations using single-precision floating-point operations. This option not only increases throughput but also reduces memory requirements, making the combined processing an additional ~1.7x faster. With these updates, performing DSR on the VNC data directly is feasible on a system with 128 GB of RAM.

We acknowledge that bugs may arise when using the software on hardware configurations with fewer resources from those on which it was developed. This was the case we noticed in the reviewer's stitching code review. While the stitching process completed, there was an issue in the MIP generation pipeline that only affected systems with less than 256 GB of RAM, where the software tried to load the entire stitched volume to RAM. To resolve this issue, we implemented a check for the total system RAM and to automatically use batch processing if the stitched data size exceeds half of the system RAM.

We are committed to improving the documentation based on user. Users can bring software bugs to our attention by submitting a GitHub issue.

Version 3:

Decision Letter:

16th Sep 2024

Dear Gokul,

I am pleased to inform you that your Article, "Image processing tools for petabyte-scale light sheet microscopy data", has now been accepted for publication in Nature Methods. The received and accepted dates will be Feb 15, 2024 and Sep 16, 2024. This note is intended to let you know what to expect from us over the next month or so, and to let you know where to address any further questions.

Over the next few weeks, your paper will be copyedited to ensure that it conforms to Nature Methods style. Once your paper is typeset, you will receive an email with a link to choose the appropriate publishing options for your paper and our Author Services team will be in touch regarding any additional information that may be required. It is extremely important that you let us know now whether you will be difficult to contact over the next month. If this is the case, we ask that you send us the contact information (email, phone and fax) of someone who will be able to check the proofs and deal with any last-minute problems.

Please note that *Nature Methods* is a Transformative Journal (TJ). Authors may publish their research with us through the traditional subscription access route or make their paper immediately open access through payment of an article-processing charge (APC). Authors will not be required to make a final decision about access to their article until it has been accepted. Find out more about Transformative Journals

Authors may need to take specific actions to achieve compliance with funder and institutional open access mandates. If your research is supported by a funder that requires immediate open access (e.g. according to Plan S principles) then you should select the gold OA route, and we will direct you to the compliant route where possible. For authors selecting the subscription publication route, the journal's standard licensing terms will need to be accepted, including self-archiving policies. Those licensing terms will supersede any other terms that the author or any third party may assert apply to any version of the manuscript.

If you are active on Twitter/X, please e-mail me your and your coauthors' handles so that we may tag you when the paper is published.

Best regards,
Rita

Rita Strack, Ph.D.
Senior Editor
Nature Methods

Visit the Springer Nature Editorial and Publishing website at http://editorial-jobs.springernature.com?utm_source=ejp_NMeth_email&utm_medium=ejp_NMeth_email&utm_campaign=ejp_NMeth for more information about our career opportunities. If you have any questions please click [here](mailto:editorial.publishing.jobs@springernature.com).
